# TWEAK/Fn14 signalling driven super-enhancer reprogramming promotes pro-metastatic metabolic rewiring in triple-negative breast cancer

Nicholas Sim [1], Jean-Michel Carter [1], Kamalakshi Deka[1], Benita Kiat Tee Tan[2,3,4], Yirong Sim[2,3,4], Suet-Mien Tan [1] & Yinghui Li [1] ✉

Triple Negative Breast Cancer (TNBC) is the most aggressive breast cancer subtype suffering from limited targeted treatment options. Following recent reports correlating Fibroblast growth factor-inducible 14 (Fn14) receptor overexpression in Estrogen Receptor (ER)-negative breast cancers with metastatic events, we show that Fn14 is specifically overexpressed in TNBC patients and associated with poor survival. We demonstrate that constitutive Fn14 signalling rewires the transcriptomic and epigenomic landscape of TNBC, leading to enhanced tumour growth and metastasis. We further illustrate that such mechanisms activate TNBC-specific super enhancers (SE) to drive the transcriptional activation of cancer dependency genes via chromatin looping. In particular, we uncover the SE-driven upregulation of Nicotinamide phosphoribosyltransferase (NAMPT), which promotes NAD+ and ATP metabolic reprogramming critical for filopodia formation and metastasis. Collectively, our study details the complex mechanistic link between TWEAK/Fn14 signalling and TNBC metastasis, which reveals several vulnerabilities which could be pursued for the targeted treatment of TNBC patients.

Breast cancer (BC) is the leading malignancy in women (which comprises 11.7% of total cases), accounting for 6.9% of all cancer related deaths globally[1-4]. Additionally, the heterogeneity of this disease makes it challenging for both diagnosis and treatment[5,6]. There are five distinct molecular subtypes of BC: Luminal A, Luminal B, HER2-enriched, Basal-like/Triple negative and Normal-like[7-9]. Amongst them, TNBC is the most aggressive and heterogenous subtype that exhibits enhanced proliferative and metastatic capacity, poorer prognosis and higher disease recurrence compared to the other subtypes. In addition, TNBCs are insensitive to endocrine and HER2-targeted therapy due to the absence of all three hormonal receptors which limits

treatment options to standard chemotherapeutic regimens, such as taxanes, anthracyclines and platinum-based agents, alongside recent combination treatment with immuno-therapeutics[10-12]. However, despite various treatment options, fewer than 30% of patients achieve pathologic complete response[10,13]. Consequently, there is an urgent need to identify effective molecular markers or driver factors specific to TNBC patients for targeted therapies.

To discover new therapeutic targets, it is crucial to elucidate the oncogenic signalling mechanisms and their requisite gene regulatory programmes which sustain TNBC malignancy. The Tumour necrosis factor (TNF)-like weak inducer of apoptosis (TWEAK)/Fibroblast

[1]School of Biological Sciences (SBS), Nanyang Technological University (NTU), 60 Nanyang Drive, Singapore 637551, Singapore. [2]Division of Surgery and Surgical Oncology, Department of Breast Surgery, National Cancer Centre Singapore, 30 Hospital Blvd, Singapore 168583, Singapore. [3]Division of Surgery and Surgical Oncology, Department of Breast Surgery, Singapore General Hospital, 31 Third Hospital Ave, Singapore 168753, Singapore. [4]SingHealth Duke-NUS Breast Centre, Singapore, Singapore. ✉e-mail: liyh@ntu.edu.sg

growth factor-inducible 14 (Fn14) pathway is one such signalling cascade implicated in the pathogenesis of TNBC and other solid tumours. The TWEAK cytokine, TNFSF12, is a member of the TNF superfamily that is widely expressed in several tissues and cell types including fibroblasts, immune, mesenchymal, endothelial and tumour cells[14–18]. Through binding to its cognate receptor, Fn14 (or TNFRSF12A), TWEAK activates multiple cellular responses including modulation of cell death, proliferation, migration, angiogenesis and production of proinflammatory mediators[19,20]. While TWEAK and Fn14 expression are typically low in most normal tissues, their aberrant expression has been detected during tissue injury or chronic inflammation, and in various tumours and metastases such as gliomas, prostate and BCs[21–25]. Notably, TWEAK and Fn14 have been found to be overexpressed in ER-negative BCs relative to the ER-positive subtype and activation of this pathway has been further correlated with increased breast tumour invasion and metastasis[26].

Aberrant TWEAK/Fn14 signalling has been shown to activate several pro-oncogenic pathways such as NF-κB, JNK, ERK and TRAF signalling[27–30]. Many of the transcription factors activated downstream of these pathways have been reported to mediate tumour cell invasion, metastasis and cancer stem cell phenotypes in multiple cancer types[31–35]. Additionally, some factors such as those from the AP-1 or NF-κB family are known to regulate the epigenome in various disease or inflammatory settings[36–40]. However, the clinical significance and biological roles of deregulated TWEAK/Fn14 signalling in regulating the epigenetic plasticity and transcriptome of TNBCs remain elusive. In this study, we performed integrative transcriptomic and epigenomic analyses of TNBC cell lines and patient samples to characterise the gene regulatory roles of TWEAK/Fn14 pathway in remodelling the SE landscape of TNBC tumours. We identify a distinct TWEAK-induced transcriptional signature in basal-like tumours that is enriched in pro-metastatic and metabolic genes. We further demonstrate that constitutive TWEAK/Fn14 signalling activates a significant proportion of TNBC-specific SEs targeting several of these genes through chromatin interactions. Importantly, deleting one of the TNBC-specific SEs targeting a TNBC dependency gene, *NAMPT*, abrogated the TWEAK/Fn14-driven tumorigenesis and metastasis of TNBC xenografts. Moreover, TWEAK/Fn14-induced NAMPT expression resulted in nicotinamide adenine dinucleotide (NAD) metabolic rewiring that is critical for the filopodia formation and invasion of TNBC cells. Collectively, our findings reveal a previously unrecognised role for TWEAK/Fn14 pathway in reprogramming the SE landscape of TNBC tumours to drive the expression of metastasis and metabolic genes critical for TNBC progression. Our work thereby highlights the therapeutic potential of targeting the TWEAK/Fn14 signalling cascade and future characterisation of the oncogenic functions of its downstream gene targets for plausible intervention.

## Results

### Persistent Fn14 activation drives TNBC tumorigenesis and metastasis

The expression of *Fn14* in breast tumours has previously been shown to be positively correlated with HER2 expression as well as the lack of ER status[26,41]. To evaluate the prevalence of aberrant TWEAK/Fn14 signalling in patients with TNBC, we interrogated the TCGA BRCA RNA-seq dataset which revealed the overexpression of *Fn14* in basal-like tumours, the predominant molecular subtype of TNBC, relative to ER-positive tumours (Fig. 1a). Further analysis of primary TNBC patient samples demonstrated higher Fn14 protein expression levels in TNBC tumours relative to their adjacent normal tissues (6 out of 9 malignant samples) (Fig. 1b). The elevated expression of Fn14 was also consistent in TNBC cell lines compared to ER-positive cell lines. Surprisingly, HER2 cell lines exhibited low Fn14 protein expression (Fig. 1c), suggesting a possible unreported mechanism for the post-transcriptional regulation of Fn14 in the HER2 subtype. Further single cell RNA-seq

(scRNA-seq) analysis of *TWEAK* and *Fn14* expression in a TNBC primary tumour and its matched lymph node metastases revealed that in both primary and metastatic settings, *Fn14* was most highly expressed in epithelial cells. In contrast, *TWEAK* was most highly expressed in endothelial cells in the primary tumour and in monocytes and macrophages in the lymph node metastases[42] (Supplementary Fig. 1). Together, this suggests that in TNBCs, TWEAK/Fn14 activation occurs through a paracrine mechanism.

Notably, *Fn14* overexpression was selectively associated with a poorer relapse-free survival in TNBC and HER2 patients but not in patients with ER-positive BCs (Supplementary Fig. 2), suggesting that TWEAK/Fn14 signalling may trigger subtype specific responses in BC patients. These observations therefore recapitulate the predominant overexpression of Fn14 in TNBC tissues relative to ER-positive subtypes and normal breast tissues, implicating the critical involvement of Fn14 signalling in the pathogenesis of TNBCs.

To functionally validate this, we treated a panel of TNBC cell lines - MDA-MB-231, Hs578T and BT549, HER2 cell lines - SKBR3 and MDA-MB-453, and ER-positive BC cell lines - MCF7, T47D and CAMA1, with exogenous TWEAK ligand. Only the TWEAK-induced TNBC cell lines displayed a striking increase in cancer cell invasion and proliferation (Fig. 1d–f). These effects were abolished upon CRISPR-mediated *Fn14* deletion in TNBC cells, confirming that the oncogenic activity of TWEAK/Fn14 signalling is driven through specific binding of TWEAK to the Fn14 receptor (Supplementary Fig. 3a–d). Consistent with the functional effects observed in vitro, TWEAK-overexpressing MDA-MB-231 cells, but not TWEAK-overexpressing MCF7 cells, displayed enhanced tumour growth and metastasis in orthotopic tumour xenografts (Fig. 1g–i, Supplementary Fig. 4). We further evaluated the impact of the tumour microenvironment on TWEAK/Fn14-driven TNBC progression. Here, we found that *Fn14* knockdown (KD) reduced the tumour growth and metastasis of 4T1 cells (mouse TNBC cell line) in immune competent BALB/cN mice (Supplementary Fig. 5). Collectively, these findings highlight the causal role of constitutive TWEAK/Fn14 activation in promoting TNBC malignancy, which is likely to confer a worse clinical outcome in TNBC patients.

### TWEAK/Fn14 signalling induces distinct TNBC transcriptional signatures associated with pro-metastatic and metabolic programmes

To examine the gene regulatory programmes mediating the distinct phenotypic effects induced by TWEAK/Fn14-signalling among BC subtypes, we performed RNA-seq analyses of the changes in their transcriptional regulome upon Fn14 activation. Intriguingly, constitutive TWEAK/Fn14 signalling induced distinct transcriptomic profiles in TNBC versus HER2 and ER-positive cell lines (Fig. 2a, Supplementary Fig. 6a, b). We identified 220 upregulated and 118 downregulated differentially regulated genes (DEGs) in TNBC cell lines that were also selectively enriched and depleted, respectively, in basal-like tumours compared to other subtypes from the TCGA-BRCA dataset (Fig. 2b). Remarkably, these genes were prominently enriched in pro-metastatic, proliferative and metabolic functions (Fig. 2c), reaffirming the oncogenic effects and clinical significance of aberrant TWEAK/Fn14 signalling in TNBC progression (Fig. 1). In contrast, TWEAK-induced DEGs in HER2 and ER-positive cell lines displayed a low concordance with the transcriptomic profiles of their respective tumour samples (Supplementary Fig. 6c, d), suggesting the oncogenic activity of the TWEAK/Fn14 pathway may be subtype specific.

Further investigation of the biochemical pathways activated downstream of TWEAK/Fn14 signalling revealed a similar enrichment of MAPK, AP-1, TNF and NF-κB pathways in all three subtypes, consistent with previous literature[43–46]. Interestingly, we also observed subtype-specific pathway enrichment such as pathways associated with NAD+ biosynthesis, methionine and purine metabolism, induction of JNK and cJun phosphorylation in TNBC cell lines (Fig. 2d).

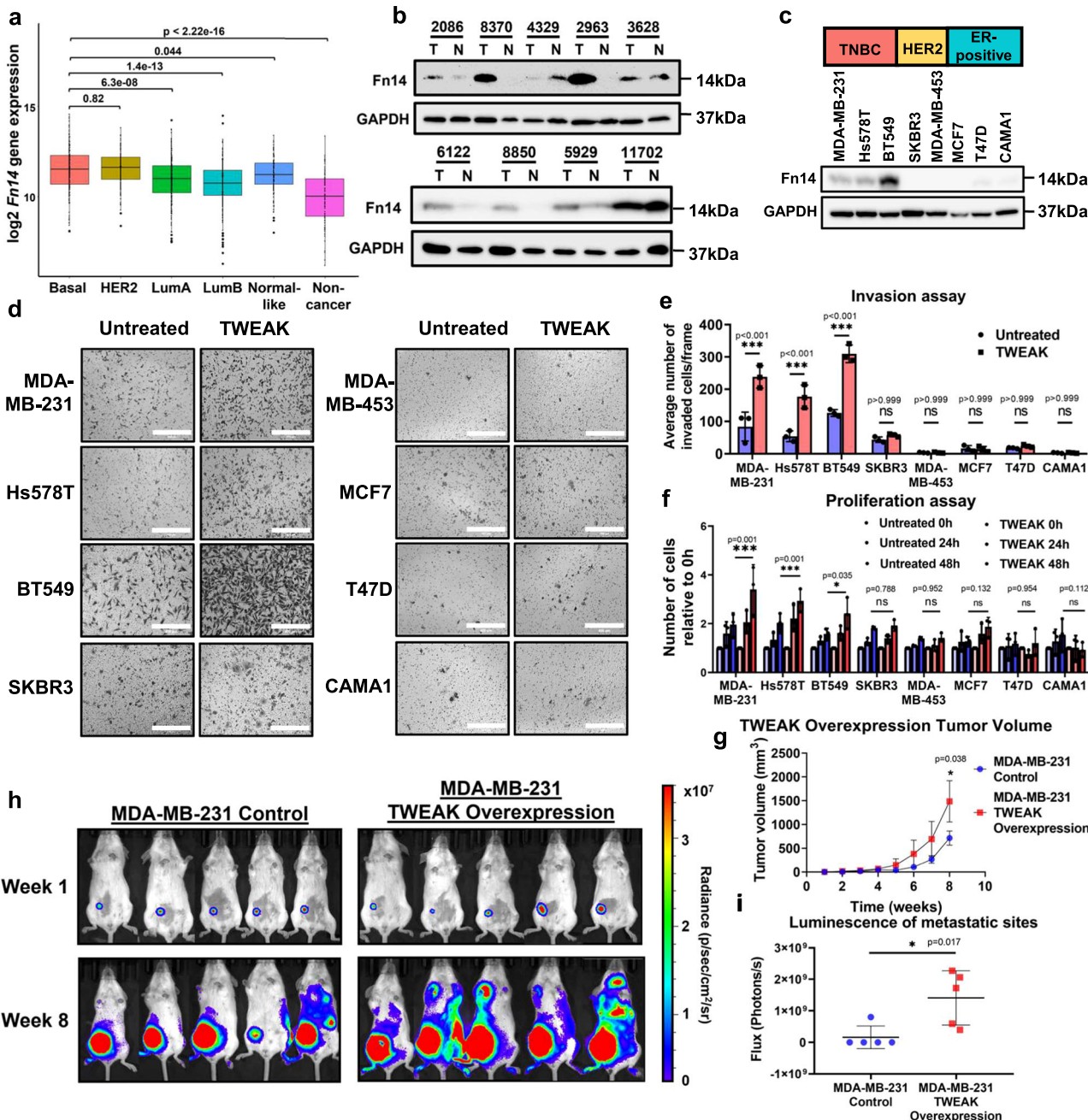

**Fig. 1 | Constitutive Fn14 expression correlates with poor survival in TNBC patients and drives TNBC progression. a** Relative *Fn14* gene expression levels in Basal-like (*n* = 229), HER2 (*n* = 158), Luminal A (*n* = 300), Luminal B (*n* = 304), Normal-like (*n* = 106) and non-cancer (*n* = 113) patient samples from the TCGA BRCA RNA-seq dataset. Box plot depicts the first quartile, median and third quartile of values. Fn14 protein expression analysed by western blotting in (**b**) TNBC patient tumours (T) and their matched normal samples (N), and in (**c**) TNBC, HER2 and ER-positive breast cancer cell lines. **d** Transwell invasion assay was performed in MDA-MB-231, Hs578T, BT549, SKBR3, MDA-MB-453, MCF7, T47D and CAMA1 cells with and without TWEAK treatment. Representative images from *n* = 3 biological replicates are shown. Scale bars: 400 μm. **e** Plot depicts the average number of invaded cells/frame (mean ± s.d) from *n* = 3 biological replicates, across four fields per replicate. **f** Proliferation assay plot depicts the average number of cells counted relative to 0 h (mean ± s.d) in MDA-MB-231, Hs578T, BT549, SKBR3, MDA-MB-453, MCF7, T47D and CAMA1 cells, treated with and without TWEAK from *n* = 3

biological replicates. **g** Tumour growth plot depicts the weekly average tumour volume (mean ± s.d) from mice injected with MDA-MB-231 cells overexpressing luciferase and overexpression control or TWEAK. Performed in *n* = 5 biological replicates. **h** IVIS tracking of mice injected with MDA-MB-231 cells overexpressing luciferase and overexpression control or TWEAK. Representative bioluminescent images of the animals were taken at 1 and 8 weeks after orthotopic xenograft. **i** Plot depicts total flux (mean ± s.d) at the metastatic sites of each animal after 8 weeks in *n* = 5 biological replicates. Western blot samples were derived from the same experiment and on the same gels for Fn14 and GAPDH. Experiments involving cell lines were performed 3 times independently, each time on different days. Two-sided Wilcoxon signed-rank test was used for differential gene expression. Two-sided two-way ANOVA was used for in vitro proliferation and invasion assay. Two-sided *t* test was used for in vivo assays. *$P < 0.05$; **$P < 0.01$; ***$P < 0.001$. Source data are provided as a Source Data file.

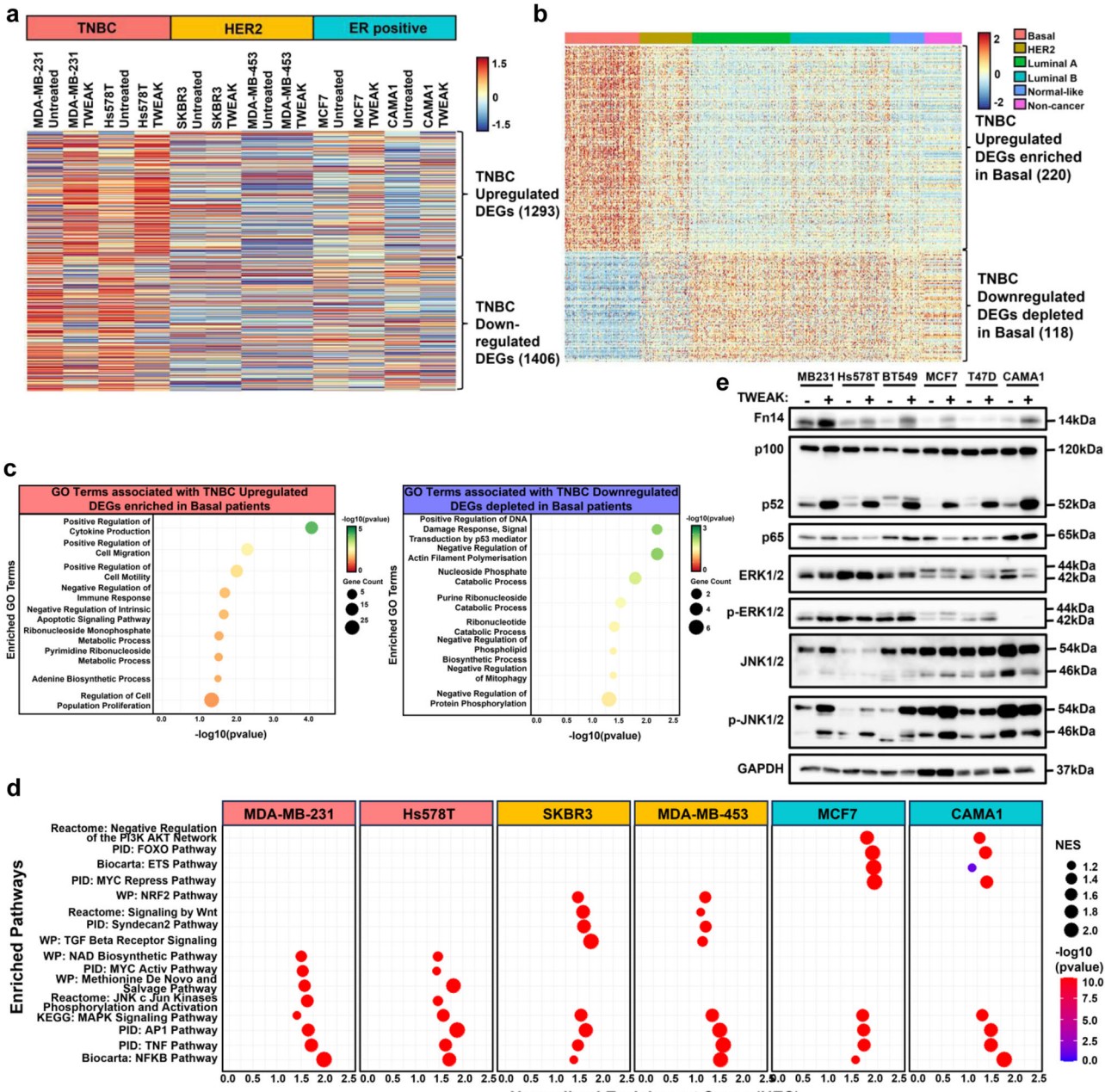

**Fig. 2 | TWEAK/Fn14 signalling drives a distinct transcriptomic signature which is associated with oncogenic processes in TNBC tumours. a** Heatmap depicting the average expression of TWEAK/Fn14 differentially regulated TNBC genes in MDA-MB-231, Hs578T, SKBR3, MDA-MB-453, MCF7 and CAMA1 cell lines, treated with and without TWEAK. Data shown represent $n = 3$ biological replicates. **b** Heatmap depicting the expression of TWEAK/Fn14 upregulated and down-regulated genes in TNBC cell lines that were respectively enriched and depleted in TCGA BRCA basal-like patient samples. **c** Dot plot depicting the enriched GO terms associated with the TWEAK/Fn14 differentially regulated genes in TNBC cell lines and TCGA BRCA basal-like patient samples. **d** Dot plot depicting enriched signalling pathways specific to and common between TNBC, HER2 and ER-positive breast cancer cell lines following TWEAK treatment. **e** NF-κB and MAPK pathway signalling regulators analysed by western blotting in MDA-MB-231, Hs578T, BT549, MCF7, T47D and CAMA1 cells, treated with and without TWEAK. The western blot samples derive from the same experiment but different gels for Fn14, another for NFKB2, p65 and GAPDH, another for p-JNK1/2 and JNK1/2 and another for p-ERK1/2 and ERK1/2 were processed in parallel. Fisher's exact test was used for statistical analysis of GO terms. Kolmogorov-Smirnov test was used for statistical analysis of enriched pathways. Source data are provided as a Source Data file.

Consistent with earlier survival analyses (Supplementary Fig. 2), we observed that the upregulation of genes associated with TNBC or HER2-specific pathways conferred Basal and HER2 patients, respectively, to poorer relapse-free survival. In contrast, high expression of genes associated with ER-positive specific pathways improved the relapse-free survival of Luminal A and Luminal B patients (Supplementary Fig. 6e), further suggesting the subtype specificity of TWEAK/Fn14-driven oncogenicity.

Immunoblotting analysis of control versus TWEAK-treated TNBC, HER2 and ER-positive BC cell lines validated the similar levels of non-canonical NF-κB activation, without impacting canonical NF-κB, in all cell lines following TWEAK stimulation. Furthermore, in line with our pathway analyses, TWEAK stimulation resulted in the selective induction of JNK signalling in TNBC cell lines (Fig. 2e, Supplementary Fig. 6f). However, ERK phosphorylation, a major signalling event downstream of MAPK activation, was not affected by TWEAK stimulation. The

activation of JNK and non-canonical NF-κB pathways was abolished in TWEAK-treated Fn14 knockout (KO) TNBC cells (Supplementary Fig. 3a), along with the pro-tumorigenic effects of TWEAK/Fn14 signalling (Supplementary Fig. 3b–d). We further demonstrate that JNK and non-canonical NF-κB activation in TWEAK-treated TNBC cell lines was ameliorated following 96 h wash-off (Supplementary Fig. 7a), indicating the requirement for persistent TWEAK/Fn14 activation to drive these pathways. Importantly, these pathways have been shown to be critical mediators of TNBC progression via the activation of oncogenic transcription factors including AP-1[33]. Our observations thereby highlight the distinct regulation of pro-metastatic and metabolic transcriptional programmes by TWEAK/Fn14 pathway in TNBCs, which are likely to be mediated via AP-1, non-canonical NF-κB and JNK signalling.

### TWEAK/Fn14 signalling stimulates oncogenic super-enhancer activation in TNBCs

Several of the TWEAK/Fn14-activated transcription factors (TFs) such as NF-κB2/p52 and AP-1 can bind accessible chromatin regions to modulate the genomic landscape of cancers and drive aberrant transcription[36,46–48]. In particular, these factors have been reported to coordinate enhancer function and various epigenomic alterations to potentiate transcription in different diseases and cell types[36,39,49]. To gain mechanistic insights into the TWEAK/Fn14-driven chromatin dynamics and *cis*-regulatory changes that impact oncogenic transcription, we performed ATAC-seq and H3K27ac ChIP-seq in our panel of BC cell lines and TNBC patient samples. Notably, TWEAK/Fn14 activation resulted in the gain and loss of chromatin accessibility at 4186 and 550 genomic loci, respectively, in TNBC cell lines (Fig. 3a). In contrast, these sites were generally unperturbed in the HER2 cell line, SKBR3, and ER-positive cell line, MCF7, suggesting that TWEAK/Fn14-induced chromatin changes may be subtype specific. These differential accessible regions displayed corresponding enhanced or depleted accessible chromatin elements in Fn14-high TNBC tumours (2086, 8370, 2063, 6122, 8850 and 5929) relative to matched normal tissues, implying that TWEAK/Fn14-driven chromatin remodelling may be impacting TNBC development. Conversely, TWEAK/Fn14-mediated chromatin dynamics observed in SKBR3 and MCF7 cells did not appear to be differentially regulated in TNBC tumours versus normal tissues (Supplementary Fig. 8a, b). Altogether, these data illustrate a previously unrecognised role for TWEAK/Fn14 pathway in modulating the chromatin plasticity of TNBCs, potentially to drive transcriptional programmes critical for TNBC progression.

Further analysis of the TWEAK/Fn14-altered accessible chromatin regions in TNBC revealed their predominant distribution (>90%) at intronic and intergenic regions, implicating a plausible rewiring of the enhancer landscape (Fig. 3b). Consistently, TWEAK/Fn14-induced H3K27 acetylation changes were positively correlated with the chromatin accessibility dynamics observed in TNBC cell lines (Fig. 3c). Differential H3K27ac signals in TWEAK/Fn14-activated TNBC cell lines displayed a distinct TNBC-specific signature in Fn14-high tumours relative to adjacent normal tissues (Fig. 3d). In contrast, this TWEAK/Fn14-induced TNBC enhancer landscape was weakly regulated upon TWEAK stimulation in HER2 and ER-positive cell lines (Fig. 3d), correlating with the chromatin accessibility dynamics. Conversely, TWEAK/Fn14-induced H3K27ac peaks in HER2 and ER-positive cell lines were mostly associated with enhancer activation in TNBCs (Supplementary Fig. 8c, d). Furthermore, these TNBC-specific epigenetic changes were abolished following Fn14 KO (Supplementary Fig. 3e) or 96 h wash-off (Supplementary Fig. 7b), highlighting the requirement for persistent TWEAK/Fn14 activation to drive enhancer activity. Gene ontology analysis of genes associated with these TNBC-specific enhancers revealed the significant enrichment of cancer-promoting biological processes including migration, cell proliferation and angiogenesis (Fig. 3e). Such epigenomic profiles recapitulate the distinct

transcriptomic signatures and oncogenic functions of TWEAK/Fn14 signalling highlighted previously in TNBCs (Figs. 1 and 2).

To further characterise the gene regulatory roles of TWEAK/Fn14-driven TNBC enhancers in promoting malignancy, we examined the prevalence of TWEAK/Fn14-dynamic SEs in our clinical samples and their association with oncogenic transcription. Multiple studies have identified the critical roles of TNBC-specific SEs in promoting subtype specific tumorigenic and metastatic functions[50,51], highlighting this as an exploitable therapeutic vulnerability. Interestingly, approximately 25–60% of all SEs detected in TNBC cell lines were differentially regulated upon TWEAK/Fn14 activation and they displayed substantial concordance between the two TNBC cell lines (Supplementary Fig. 9). A large proportion (~70%) of these TWEAK/Fn14-dynamic SEs were also detected in Fn14-high TNBC tumours (Fig. 3f). Specifically, TWEAK-gained SEs exhibited elevated enhancer activity in tumours relative to adjacent normal tissues, which correlated with a significantly higher average relative expression of neighbouring genes in basal-like tumours compared to other BC subtypes and non-cancer tissues (Fig. 3g, h). These findings highlight the distinct regulation of TWEAK/Fn14 pathway in the reprogramming of TNBC-specific SEs, which are clinically important and likely to drive the aberrant expression of oncogenes essential for TNBC progression.

### TWEAK/Fn14-activated oncogenic SEs are enriched with AP-1 binding sites

Next, we sought to identify the DNA binding factors associated with the TWEAK/Fn14-driven TNBC SE landscape by performing ATAC-seq footprinting analyses in the TNBC cell lines and clinical samples. Our analyses revealed the enhanced enrichment of NF-κB and AP-1 TF footprints genome-wide in both TWEAK/Fn14 activated TNBC cell lines and Fn14-high TNBC tumours (Fig. 4a, b, Supplementary Fig. 10). This is consistent with the induction of NF-κB and MAPK pathways in TWEAK/Fn14-activated TNBC cells (Fig. 2d, e) and consequent activation of NF-κB and AP-1 TFs that have been implicated in disease-associated enhancer/SE regulation[36–40,48]. Notably, a cluster of AP-1 TF footprints exhibited the highest enrichment in Fn14-high tumours, suggesting an association between increased AP-1 DNA binding with TNBC development and TWEAK/Fn14 activation. Intriguingly, further analysis demonstrated increased binding activity of many AP-1 TFs in Fn14-high TNBC tumours specifically at TWEAK/Fn14-driven TNBC SEs (Fig. 4c). Furthermore, the genes proximal to these enriched SE binding sites were mainly associated with cell proliferation, migration and nucleotide metabolism, mirroring the cancer-promoting processes enriched in the TWEAK-induced transcriptomic and enhancer signatures (Fig. 4d). These observations implicate the enhanced AP-1 binding activities in TNBC during TWEAK/Fn14-driven epigenomic alterations and oncogenesis. We functionally verified this through supplementation of an AP-1 inhibitor, T5224, which abolished TWEAK/Fn14-driven proliferation and invasion in MDA-MB-231 and Hs578T cells (Supplementary Fig. 11a–c). Consistent with the in vitro data, T5224 supplementation reduced the TWEAK/Fn14-driven metastasis of TNBC cells in vivo without affecting tumour growth (Supplementary Fig. 11d–g). Our data therefore suggest that AP-1 factors are critical regulators of TWEAK/Fn14-induced oncogenic SEs and function.

### TWEAK/Fn14 activation drives enhancer-promoter looping associated with TNBC dependency genes

Distal enhancers and SEs have been shown to activate the transcription of cancer driver genes through enhancer-promoter *cis*-regulatory interactions in various malignancies including TNBCs[52–54]. To determine whether the TWEAK/Fn14-dynamic, TNBC-specific SEs were interacting with potential oncogenes to enhance malignancy, we performed H3K27ac HiChIP to characterise the chromatin interactome in control versus TWEAK-treated MDA-MB-231 cells. TWEAK/Fn14 activation induced a drastic remodelling of the TNBC chromatin architecture,

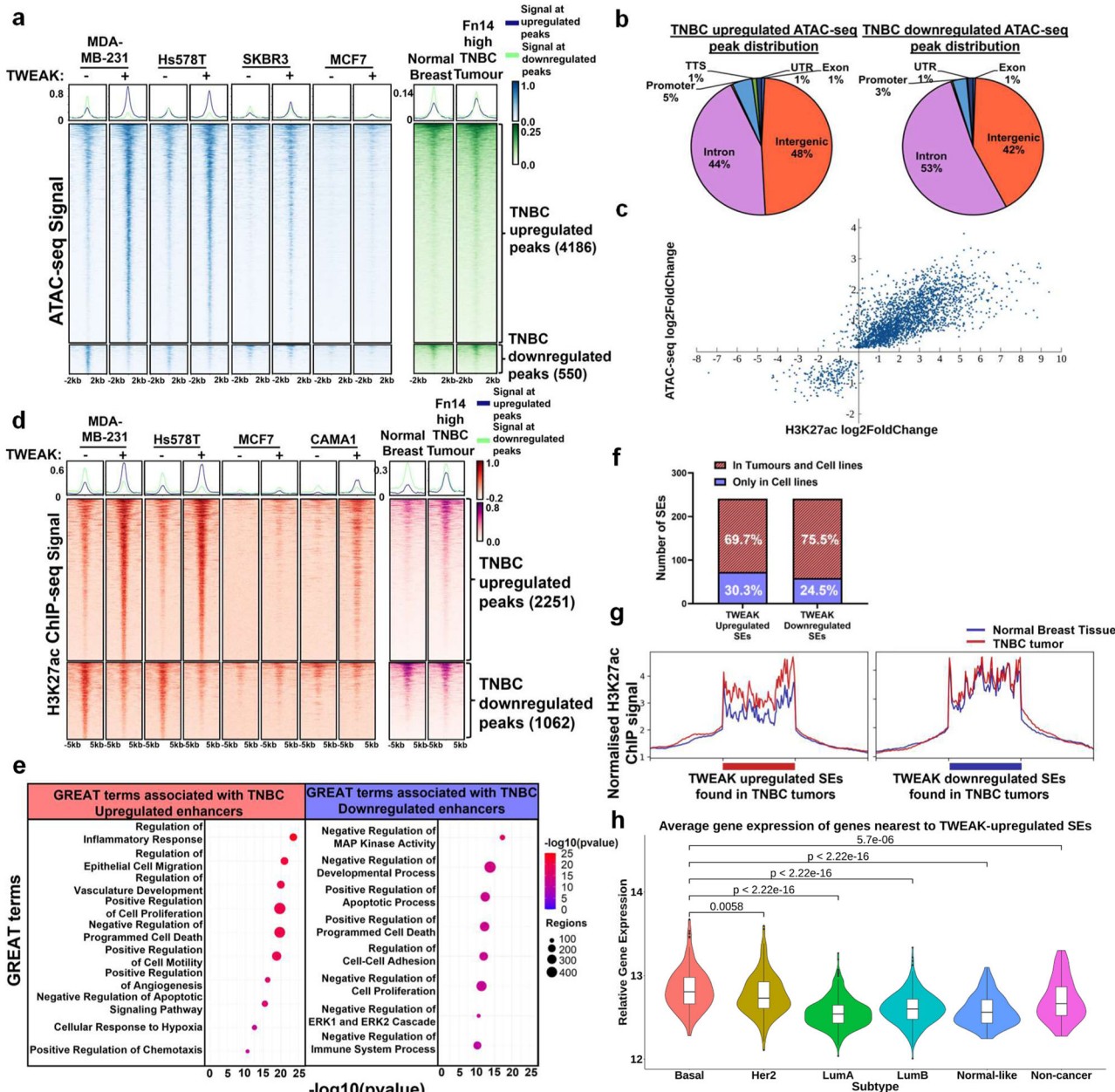

**Fig. 3 | TWEAK/Fn14 signalling rewires the SE landscape of TNBCs. a** ATAC-seq signals of MDA-MB-231, Hs578T, SKBR3 and MCF7 cells, treated with and without TWEAK, as well as the merged signals of Fn14 high TNBC patient tumours and their matched normal samples (2086, 8370, 2963, 6122, 8850 and 5929) at differentially regulated chromatin accessible sites in TNBC cell lines following TWEAK treatment. **b** Genomic distribution of the TWEAK/Fn14 differentially regulated TNBC ATAC-seq peaks. **c** Correlation plot between TWEAK/Fn14 differentially regulated TNBC ATAC-seq peaks and the H3K27ac ChIP-seq peaks that overlap. **d** H3K27ac ChIP-seq signals of MDA-MB-231, Hs578T, SKBR3, MDA-MB-453, MCF7 and CAMA1 cells, treated with and without TWEAK, as well as the merged signals of Fn14 high TNBC patient tumours and their matched normal samples at differentially regulated enhancer sites in TNBC cell lines. **e** GREAT terms associated with TNBC TWEAK/ Fn14-upregulated and downregulated enhancers. **f** Bar plot depicting the number of shared TWEAK/Fn14 regulated TNBC SEs found in cell lines and tumours. **g** Merged H3K27ac normalised binding score of the Fn14 high TNBC patient tumours and their matched normal samples at TWEAK/Fn14 differentially regulated SEs found in TNBC cell lines and tumour samples. **h** Average gene expression in TCGA BRCA patient samples of the nearest genes to TWEAK/Fn14 upregulated SEs detected in TNBC cell lines and tumour samples. Basal-like ($n = 229$), HER2 ($n = 158$), Luminal A ($n = 300$), Luminal B ($n = 304$), Normal-like ($n = 106$) and non-cancer ($n = 113$). Box within violin plot depicts the first quartile, median and third quartile of values. Binomial test was used for statistical analysis of GREAT GO terms. Two-sided Wilcoxon signed-rank test was used for statistical analysis of differential gene expression. *$P < 0.05$; **$P < 0.01$; ***$P < 0.001$.

whereby 15,293 gained and 5258 lost chromatin interactions were detected from 26,011 significant chromatin loops (>10 KB to 2 MB) called across both conditions (Fig. 5a, b). We subsequently classified the loops into three annotations: E-P: Enhancer-Promoter, P-P: Promoter-Promoter and E-E: Enhancer-Enhancer. Intriguingly, majority of the differential loops, particularly gained loops, were E-P interactions (10828 loops, 41.6%) followed by E-E (9018 loops, 34.7%) and P-P (6165 loops, 23.7%) interactions (Fig. 5c). Such observations uncovered a distinct role for TWEAK/Fn14 pathway in mediating active chromatin interactions between enhancers and gene promoters in TNBC.

Further integration of the TWEAK/Fn14-induced E-P interaction dynamics with the enhancer and gene expression changes revealed a positive correlation between the *cis*-regulatory interaction dynamics and enhancer/gene expression fold change (Fig. 5d). Among the E-P

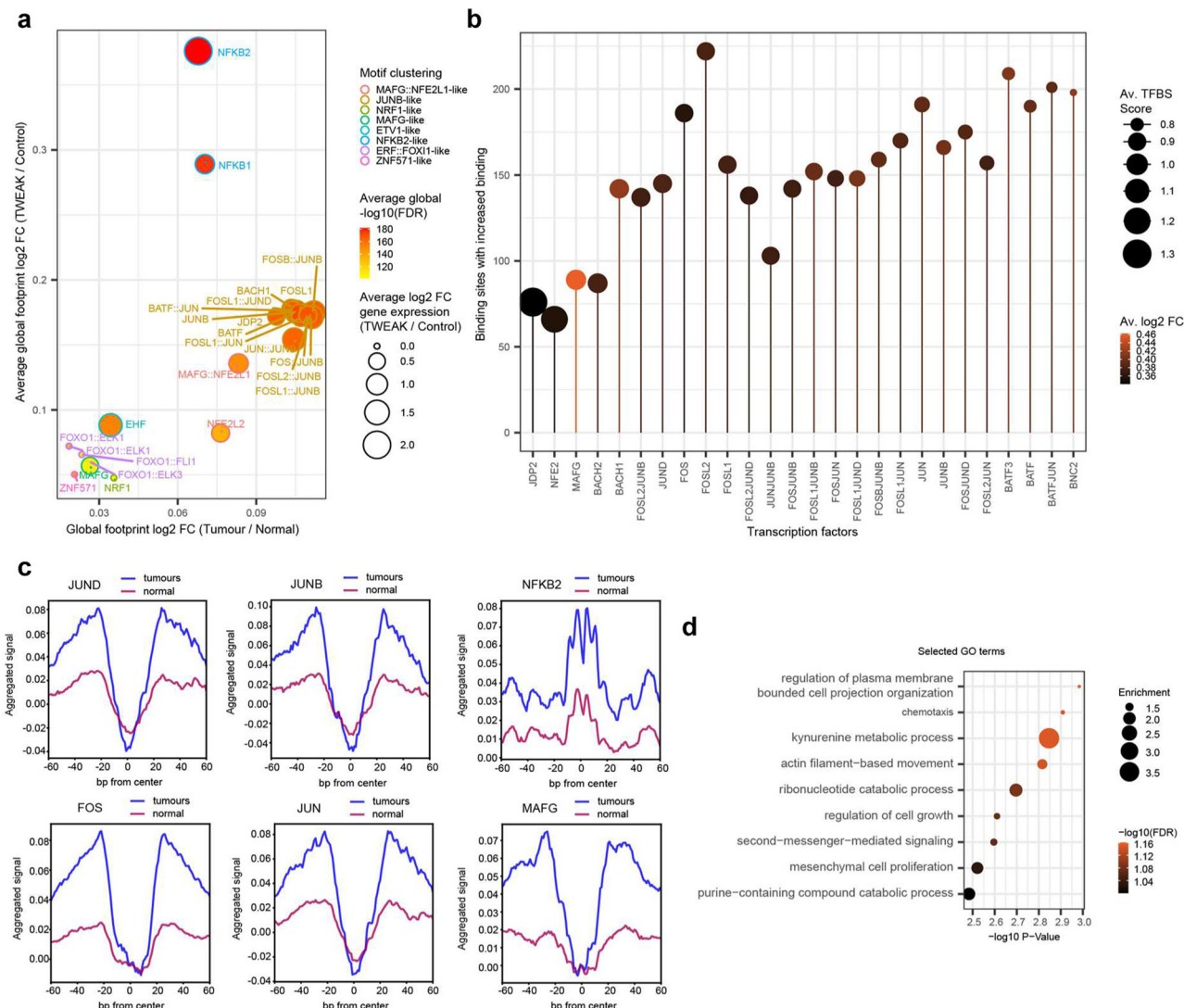

**Fig. 4 | TWEAK/Fn14 signalling alters the binding dynamics of key transcription factors from the AP-1 and NF-κB families. a** Global TF binding dynamics in TWEAK stimulated MDA-MB-231, Hs578T and Fn14 high TNBC patient tumours (2086, 8370, 2963, 6122, 8850 and 5929). MDA-MB-231 and Hs578T log2 fold changes for footprints and gene expression are averaged. Average global FDR for each TF footprint across cell lines and tumours indicated. TF binding changes exhibiting *p* values in the upper half of the distribution are selected and filtered for consistent global footprint and gene expression dynamics across cell lines and tumours. Motifs are clustered based on similarity via TOBIAS. **b** Aggregate plots for footprints called as bound in tumour samples and unbound in normal samples. Aggregated signal is normalised for coverage, centred on the TFBS flanked by 60 bp. **c** AP-1 family transcription factor binding sites at SEs showing increased binding under TWEAK stimulation and in TNBC tumours. Average transcription factor binding site (TFBS) score is calculated across all sites showing increased binding. Average log2 fold change is the average across MDA-MB-231, Hs578T under TWEAK stimulation and TNBC versus normal samples. **d** GO biological processes enriched for genes in proximity to binding sites showing increased footprint depth across enhancers under TWEAK stimulation and in TNBC versus normal samples. False discovery rate test was used for statistical analysis of GO terms.

pairs displaying epigenomic changes, 171 interactions were associated with TNBC-specific SEs and 88 interactions were associated with TWEAK/Fn14-regulated expression of 48 unique DepMap TNBC dependency genes (44 of which were also pan cancer DepMap dependency genes). Additionally, majority of the TNBC dependency genes (26 out of 48) were transcriptionally activated following TWEAK stimulation and targeted by upregulated enhancers, consistent with the TWEAK/Fn14-induced oncogenic transcriptional signatures and SE activity observed previously. Some of these dependency genes, such as *ITGAV*, *NAMPT* and *KIF18A*, have also been reported to drive TNBC proliferation and metastasis[55–58] (Fig. 5d). Notably, we detected 15 interactions between TWEAK/Fn14-dynamic TNBC SEs and differentially regulated TNBC dependency genes, where the loop targeting *NAMPT* was the most highly upregulated. Collectively, these findings demonstrate a previously unexplored role of TWEAK/Fn14-driven

epigenomic rewiring in the promotion of TNBC malignancy through augmenting enhancer-promoter looping. Our data further implicate the critical role of aberrant TWEAK/Fn14 signalling in the activation of TNBC dependency genes via oncogenic SE acquisition.

**NAMPT is a target gene of a TNBC-specific, TWEAK-activated SE**
To validate this further, we selected the *NAMPT* locus which contained the most upregulated TNBC dependency gene associated with a TWEAK/Fn14-gained TNBC SE. Clinically, *NAMPT* is overexpressed in the basal-like subtype and its high expression confers the poorer survival of patients with TNBC but not HER2- or ER-positive BCs (Supplementary Fig. 12). Moreover, TWEAK treatment selectively induced NAMPT expression in TNBC but not HER2 or ER-positive cell lines (Fig. 6a, Supplementary Fig. 13a), correlating with the elevated levels of NAMPT in Fn14-high TNBC tumours relative to matched normal tissues

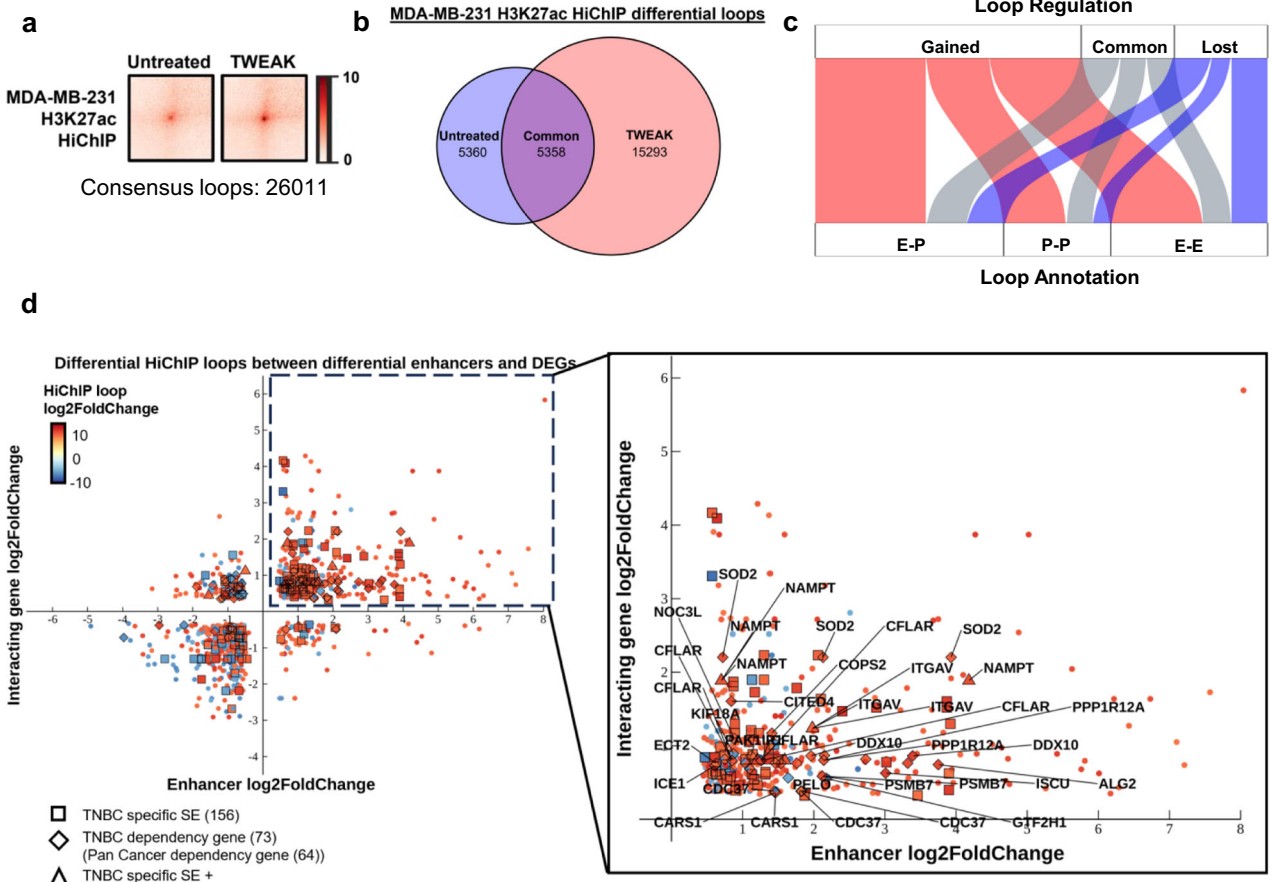

**Fig. 5 | TWEAK/Fn14 signalling remodels the TNBC chromatin interactome associated with key oncogenes. a** Aggregated peak analysis matrices depicting H3K27ac HiChIP loop signals in MDA-MB-231 cells, treated with and without TWEAK. **b** Venn diagram depicting the common and differential H3K27ac HiChIP loops between untreated and TWEAK-activated MDA-MB-231 cells. **c** Annotation of H3K27ac HiChIP loops detected in MDA-MB-231 cells, treated with and without TWEAK treatment. E-P: Enhancer-Promoter; P-P: Promoter-Promoter; E-E: Enhancer-Enhancer. **d** Plot depicting the TWEAK/Fn14 differentially regulated

HiChIP loops between enhancers and gene promoters. Squares indicate loops where the interacting enhancer overlaps a TNBC-specific SE. Diamonds indicate loops where the interacting gene promoter belongs to a DepMap TNBC dependency gene. Triangles indicate loops between a TNBC-specific SE and DepMap TNBC dependency gene. The upper right quadrant of the plot containing loops between upregulated enhancers and DEGs is enlarged, and the top 20 DepMap TNBC dependency genes are labelled.

(Fig. 6b, Supplementary Fig. 13b). These observations corroborated with the selective induction of SE activity, located ~40KB downstream of the *NAMPT* promoter, only in TNBC cells during TWEAK/Fn14 signalling (Fig. 6c).

To investigate whether this TNBC-specific SE is critical for *NAMPT* expression and TNBC development, we introduced 1 KB deletions to three constituent enhancers within the SE in MDA-MB-231 cells using CRISPR-Cas9 (Supplementary Fig. 14). Interestingly, deletion of enhancer #3, but not the other two enhancers, attenuated TWEAK-induced expression of NAMPT (Fig. 6d, Supplementary Fig. 13c). Additionally, the TWEAK-induced activity of enhancer #3 was abolished in Fn14-KO TNBC cells (Supplementary Fig. 15), demonstrating the role of this TWEAK/Fn14-driven SE in the regulation of *NAMPT* expression in TNBCs.

Further investigation of the *NAMPT* enhancer #3 deletion locus via TOBIAS footprinting analysis revealed the enhanced enrichment of AP-1 TF binding in TWEAK-activated TNBC cell lines and Fn14-high TNBC tumours (Supplementary Fig. 16a, b). This is consistent with the genome-wide footprint changes observed earlier (Fig. 4a). Intriguingly, MAFG, a poorly characterised small Maf protein from the AP-1 superfamily[59], exhibited the highest enrichment. While MAFG is expressed in all our BC cell lines, its expression was typically higher in the TNBC cell lines (Supplementary Fig. 16c). Since TWEAK-mediated

non-canonical NF-κB activation and the cooperativity of NF-κB/p52 with non-κB factors have been documented in cancer cells[43,47,60], we hypothesised the possibility of p52 facilitating MAFG recruitment to enhancer sites following TWEAK treatment. Here, we found minimal colocalisation between MAFG and p52 at differential TWEAK/Fn14-regulated TNBC enhancer sites but instead, observed increased MAFG binding at upregulated regions (Supplementary Fig. 16d). This enhanced genomic binding of MAFG at upregulated enhancer sites correlated with the increased ATAC-seq signals, further suggesting that MAFG may be recruited to the TNBC enhancers during TWEAK/Fn14-mediated upregulation of chromatin accessibility.

To evaluate the role of MAFG in mediating TWEAK-driven NAMPT expression, we found a significant positive correlation between *MAFG* and *NAMPT* expression in the TCGA BRCA and Pan-cancer dataset (Supplementary Fig. 16e). This is consistent with the MAFG and NAMPT expression levels in our cell lines (Fig. 6a, Supplementary Fig. 16c). To elucidate how MAFG could regulate NAMPT, we performed MAFG ChIP-seq in our BC cell lines. Here, within the *NAMPT* locus, we detected MAFG binding only in the TNBC cell lines at *NAMPT* enhancer #3 (Supplementary Fig. 16f). We then induced short hairpin RNA (shRNA)-mediated KD of MAFG in MDA-MB-231 and Hs578T cells. Remarkably, MAFG KD abolished TWEAK/Fn14-driven NAMPT expression and *NAMPT* enhancer #3 activation (Supplementary

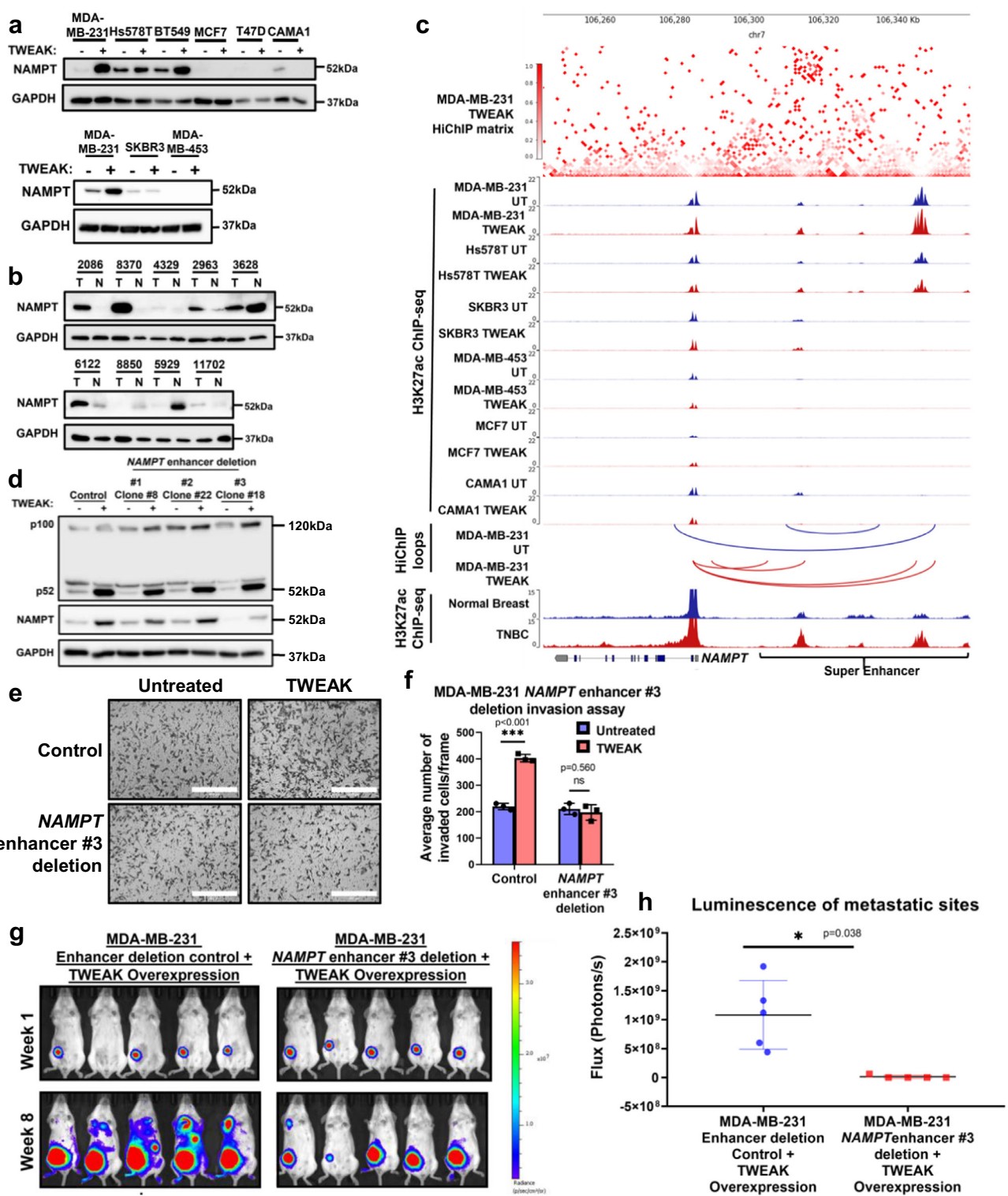

Figs. 17, 18), highlighting a previously unrecognised role of MAFG in oncogenic E-P regulation in TNBCs.

Subsequent analysis of the functional impact of *NAMPT* enhancer #3 deletion on TNBC development revealed the abrogation of TWEAK/Fn14-induced invasion, in vivo tumorigenesis and metastasis (Fig. 6e–h, Supplementary Fig. 19). KD of *NAMPT* through CRISPR-Cas9 in TNBC cells similarly attenuated these TWEAK/Fn14-driven oncogenic traits both in vitro and in vivo (Supplementary Fig. 20), further verifying that the phenotypic changes following *NAMPT* enhancer #3 deletion occurred via NAMPT. Collectively, our results highlight an

unreported regulatory mechanism of a key cancer driver gene in TNBC, that is mediated through TWEAK/Fn14 signalling. These findings depict the clinical significance of TWEAK/Fn14 pathway in reprogramming the SE landscape of TNBCs to drive oncogenic transcription and TNBC progression.

## TWEAK/Fn14-activated NAMPT expression promotes NAD+ metabolic reprogramming critical for filopodia formation

NAMPT is the rate-limiting enzyme in the nicotinamide (NAM) salvage pathway that catalyses the conversion of NAM to nicotinamide

**Fig. 6 | TWEAK/Fn14-activated TNBC SE drives NAMPT expression and TNBC progression.** Western blot analysis of NAMPT expression in (**a**) untreated and TWEAK-treated TNBC, HER2 and ER-positive breast cancer cell lines, and (**b**) TNBC patient tumours (T) and their matched normal samples (N). **c** *NAMPT* locus. Top: H3K27ac HiChIP heatmap of TWEAK-treated MDA-MB-231 cells. Middle: H3K27ac ChIP-seq of untreated and TWEAK-treated MDA-MB-231, Hs578T, SKBR3, MDA-MB-453, MCF7 and CAMA1 cells. Bottom: chromatin interactions in untreated and TWEAK-treated MDA-MB-231 cells, and merged H3K27ac ChIP-seq of Fn14 high TNBC tumours and their matched normal samples (2086, 8370, 2963, 6122, 8850 and 5929). **d** Western blot analysis of NAMPT and NFκB2 expression in Control and *NAMPT* enhancer #1, #2 and #3 deleted MDA-MB-231 cells, with and without TWEAK. **e** Transwell invasion assay performed in Control and *NAMPT* enhancer #3 deleted MDA-MB-231, cells with and without TWEAK. Scale bars: 400 μm. **f** Plot depicts the average number of invaded cells/frame (mean ± s.d), across four fields per replicate. **g** IVIS imaging of mice injected with control and *NAMPT* enhancer #3

deleted MDA-MB-231 cells overexpressing TWEAK and luciferase. Representative bioluminescent images were taken 1 and 8 weeks after injection (luminescent signal of Control mouse #2 at week 1 was detectable but very low). **h** Plot depicting the total flux (mean ± s.d) at the metastatic sites of each animal at week 8. (Representative image of Control Mouse #3 was taken at week 7 as the mouse died before week 8). The western blot samples derive from the same experiment and same gel for NAMPT and GAPDH in (**a**) and (**b**). The western blot samples derive from the same experiment but different gels for NFKB2 and GAPDH and another for NAMPT were processed in parallel in (**d**). In vivo assays represent $n = 5$ biological replicates, other assays represent $n = 3$ biological replicates. Experiments involving cell lines were performed 3 times independently, each time on different days. Two-sided $t$ test was used for in vivo assay, two-sided two-way ANOVA was used for the other assays where $*P < 0.05$; $**P < 0.01$; $***P < 0.001$. Source data are provided as a Source Data file.

mononucleotide (NMN), which is then converted to NAD+[61]. NAD+ is utilised in many essential biological processes such as ATP production which can drive cancer progression[62,63]. To investigate this, we compared the ATP levels in NAMPT high TNBC cell lines versus NAMPT low ER-positive cell lines. Here, we found that TWEAK/Fn14 signalling significantly increased NAD + /NADH and intracellular ATP levels exclusively in TNBC cell lines (Supplementary Fig. 21a, b). This is also consistent in our TNBC patient samples where Fn14 and NAMPT-high TNBC tumour samples had higher NAD + /NADH and intracellular ATP levels compared to their matched normal samples (Supplementary Fig. 21c, d). Importantly, this metabolic rewiring was NAMPT-dependent as deletion of *NAMPT* enhancer #3 or *NAMPT* KD failed to stimulate NAD + /NADH and ATP levels upon TWEAK treatment. Conversely, exogenous NAD+ supplementation was sufficient to rescue intracellular ATP levels in these TNBC cells (Fig. 7a, b, Supplementary Fig. 22a, b).

ATP is crucial for numerous biochemical functions, particularly actin polymerisation and formation of actin protrusions such as filopodia which promote cell motility[64,65]. Consistent with TWEAK-induced ATP effects, increased filopodia formation was detected in TNBC but not ER-positive cell lines following TWEAK stimulation (Supplementary Fig. 21e, f). Moreover, deletion of *NAMPT* enhancer #3 or *NAMPT* KD abolished TWEAK/Fn14-mediated filopodia formation, which was restored with the addition of NAD+ (Fig. 7c, d, Supplementary Fig. 22c, d). These NAMPT-regulated phenotypic changes correlated with the TWEAK/Fn14-induced invasiveness of TNBC cells (Fig. 7e, f, Supplementary Fig. 22e, f). Furthermore, disruption of actin polymerisation through latrunculin A (Lat A) supplementation ameliorated TWEAK or NAD + -driven filopodia formation and invasion of TNBC cells (Fig. 7c–f, Supplementary Fig. 22c–f), reaffirming the key role of TWEAK/Fn14-induced NAMPT and NAD+ metabolism in pro-metastatic functions of TNBCs. Altogether, our data establish a mechanistic link between TWEAK/Fn14-driven epigenomic alterations and metabolic rewiring in TNBC progression (Fig. 8).

## Discussion

The TWEAK/Fn14 signalling axis regulates various cellular and physiological processes including proliferation, apoptosis, migration, differentiation, angiogenesis and tissue repair during acute injury and pathological conditions. However, its precise role(s) in mediating the epigenetic switch that contributes to the dysregulated gene signature during disease pathogenesis is poorly characterised. Here, we uncovered a distinct role for this pathway in reprogramming the SE landscape of TNBCs to induce the aberrant expression of pro-metastatic and metabolic genes. We show that constitutive TWEAK/Fn14 signalling activates oncogenic SEs associated with the TNBC-specific chromatin landscape and augments the expression of TNBC dependency genes via E-P interactions. Notably, we reveal an unreported mechanism underscoring TWEAK/Fn14-induced rewiring of the

TNBC epigenome, contributing to the dysregulated expression of NAMPT critical for metastasis.

Extensive research has characterised the pivotal role of epigenetic aberrations, particularly oncogenic SE activation, in regulating the transcriptome of various cancers including BC[52], medulloblastoma[66], colorectal cancer[67,68] and multiple myeloma[69]. This often leads to dysregulated oncogenic transcription, promoting tumorigenesis, metastasis and metabolic reprogramming. Collectively, these studies demonstrate the critical role of SE-driven transcriptional dependencies in cancer malignancy and highlight potential vulnerabilities that can be exploited for therapeutic targeting. Intriguingly, our findings show that TWEAK/Fn14-activated oncogenic SEs promote aberrant gene expression programmes in TNBC through chromatin looping. In particular, we identified a TWEAK/Fn14-driven, TNBC-specific SE that is crucial for activating NAMPT expression. This in turn rewires NAD + / NADH and ATP metabolism to promote filopodia formation which is essential for TNBC metastasis.

As cancer cells develop and progress, metabolic pathways are altered to meet the increased bioenergetic demands required for tumour growth and to support cell state transition into a metastatic phenotype[70–72]. This is typically attributed to mutations in metabolic regulators such as *HIF1* and *MYC*[73,74] or the aberrant activation of pathways including PI3K and AMPK[75,76]. However, emerging evidence have implicated the critical roles of SEs in the metabolic reprogramming of cancer cells[67,77]. Here, we report a previously uncharacterised mechanism whereby constitutive TWEAK/Fn14 signalling induces the activity of an oncogenic SE targeting *NAMPT*, a key regulator of cancer cell metabolism[78,79], in TNBCs. Consequently, TWEAK/Fn14-activated NAMPT expression promoted NAD+ metabolic rewiring and the increased tumorigenesis and metastasis of TNBC cells[57,80]. These data indicate a functional link between aberrant NAD + /NADH metabolism and TNBC progression. NAD+ is intricately involved in many essential biological processes. However when dysregulated, it is linked to self-renewal and radiotherapy resistance in GBM[81], increased stemness in colon cancer and gliomas[82,83] and altered ATP metabolism which has been implicated in TNBC metastasis[84]. Despite their importance in cancer cell functions, the precise mechanism linking NAD+ and ATP metabolism to the promotion of TNBC metastasis remains elusive. Interestingly, our findings revealed that TWEAK treatment led to a NAD + -dependent increase in intracellular ATP levels selectively in TNBC cell lines. Furthermore, NAD + -dependent filopodia formation upon TWEAK/Fn14 activation correlated with the invasiveness of TNBC cell lines. These results illustrate an unrecognised mechanism wherein NAD+ drives TNBC invasion via ATP-dependent filopodia formation and highlights the complex TWEAK/Fn14 regulatory landscape in dictating TNBC progression.

Collectively, our study unveils an understudied role of TWEAK/ Fn14 signalling in the activation of oncogenic SEs to drive metabolic reprogramming and metastasis. Specifically, we reveal the mechanistic

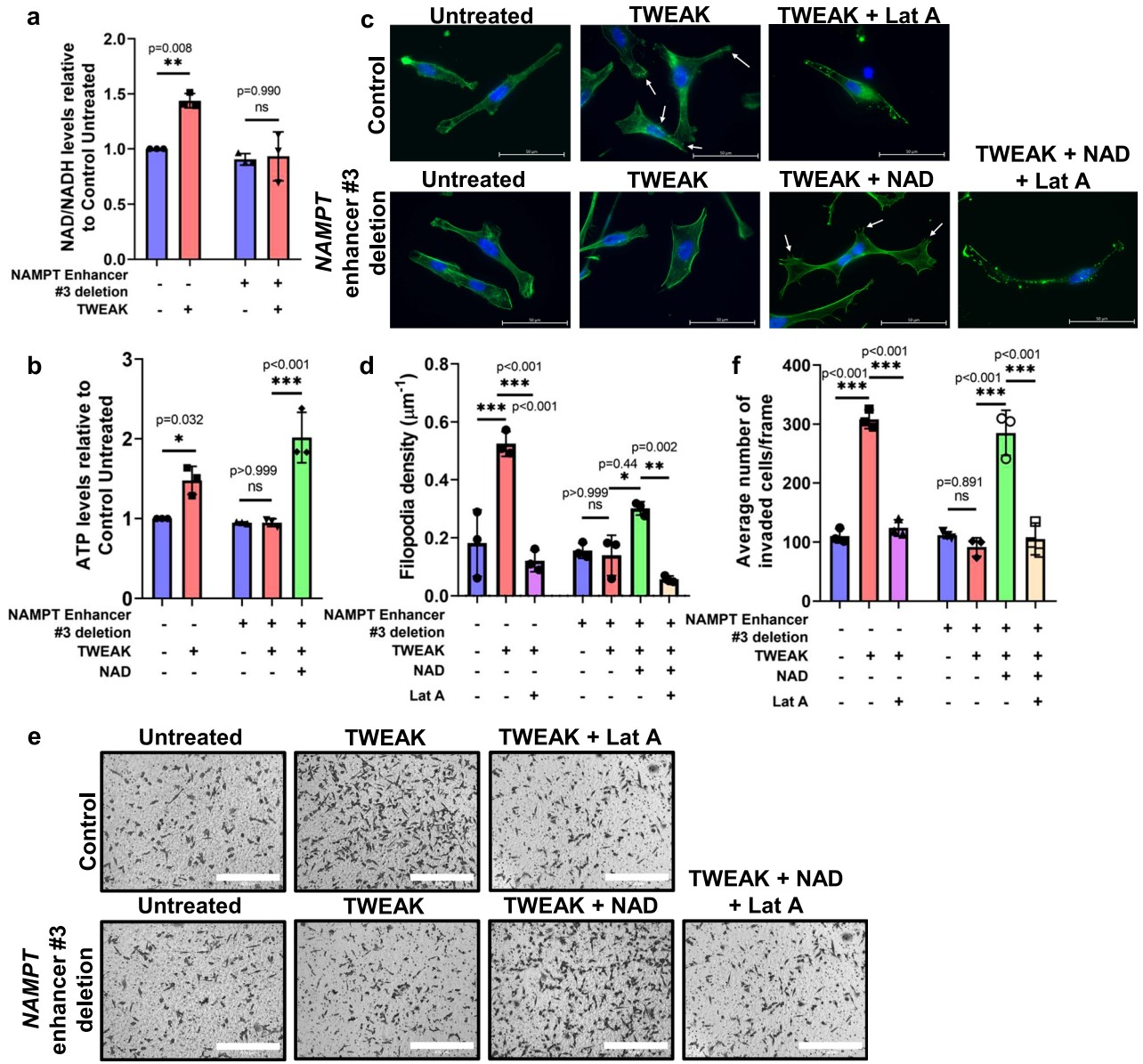

**Fig. 7 | TWEAK/Fn14-driven NAMPT regulates NAD + /NADH and ATP production to stimulate filopodia and promote invasion in TNBC cells. a** Plot depicting the relative NAD + /NADH levels in control and *NAMPT* enhancer #3 deleted MDA-MB-231 cells, with and without TWEAK treatment. **b** Plot depicting the relative intracellular ATP levels in control and *NAMPT* enhancer #3 deleted MDA-MB-231 cells, following TWEAK and NAD+ treatment. **c** Representative images of control and *NAMPT* enhancer #3 deleted MDA-MB-231 cells stained for actin and DAPI, following TWEAK, NAD+ and LatA treatment, using a Zeiss Live Cell Observer microscope (63x magnification). White arrows point to filopodia protrusions. Scale bars: 50 μm. (*n* = 3 biological replicates) (**d**) Plot depicting the average filopodia density (mean ± s.d) from *n* = 3 biological replicates. **e** Representative images of transwell invasion assay performed in Control and *NAMPT* enhancer #3 deleted MDA-MB-231 cells following TWEAK, NAD+ and LatA treatment (*n* = 3 biological replicates). Scale bars: 400 μm. **f** Plot depicts the average number of invaded cells/ frame (mean ± s.d) from *n* = 3 biological replicates, across four fields per replicate. Experiments involving cell lines were performed 3 times independently, each time on different days. Two-sided one-way ANOVA was used for statistical analysis all assays. *$P < 0.05$; **$P < 0.01$; ***$P < 0.001$. Source data are provided as a Source Data file.

basis underlying aberrant TWEAK/Fn14 activation and pro-metastatic metabolic rewiring in TNBC. These findings provide evidence for the clinical significance of TWEAK/Fn14-driven epigenomic alterations in TNBC progression, supporting further investigation of their associated transcriptional dependencies that will be useful in the discovery of effective molecular targets for TNBC therapy.

## Methods

This research complies with all relevant ethical regulations. Consent and approval for the use of clinical samples was obtained from the Nanyang Technological University Institutional Review Board,

Singapore under the protocol Oncogenic TWEAK/Fn14 driven enhancer-promoter interactions in breast cancer (IRB: IRB-2021-388), in accordance with the Human Biomedical Research Act (HBRA) requirements. All clinical samples were obtained from SingHealth Tissue Repository Tissue Bank, Singapore. Animal studies were approved by Institutional Animal Care and Use Committee of Nanyang Technological University Singapore (NTU-ARF; AUP: A21070). Tumour size/burden was monitored every 2–3 days. The maximal tumour volume approved by the IACUC was 2000 mm$^3$. Once this limit is exceeded, the mice were immediately euthanised.

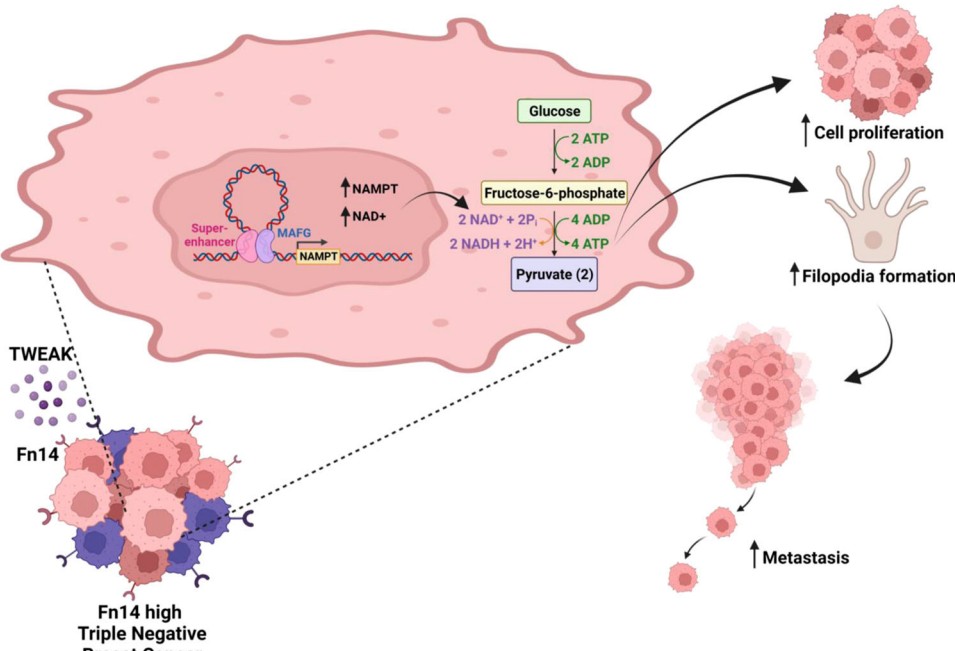

**Fig. 8 | Proposed mechanism of TWEAK/Fn14 signalling in the activation of oncogenic TNBC SEs to induce pro-metastatic metabolic reprogramming.** Aberrant TWEAK/Fn14 signalling in TNBCs trigger *NAMPT* SE activation to augment NAMPT expression via chromatin looping. MAFG binds to the TWEAK-driven SE to regulate its activity which in turn leads to NAD+ and ATP metabolic rewiring, resulting in enhanced TNBC proliferation, filopodia formation and metastasis. Created with BioRender.com.

## Cell lines and reagents

MDA-MB-231 and MCF7 BC cell lines were a gift from Asst Prof. Zhao Wenting at Nanyang Technological University, Singapore[85]. T47D BC cell line was a gift from Assoc Prof. Li Shang at Duke-NUS Medical School, Singapore. BT549, SKBR3, MDA-MB-453 and CAMA1 BC cell lines were a gift from Assoc Prof. Tam Wai Leong at the Genome Institute of Singapore, Singapore[86]. Hs578T was purchased from ATCC (HTB-126). 4T1 cells were a gift from Assoc Prof Su I-Hsin at Nanyang Technological University, Singapore. 4T1 cells were maintained in RPMI 1640 medium supplemented with 10% fetal bovine serum (FBS; Sigma). All other cell lines were maintained in Dulbecco's modified Eagle's medium (DMEM; HyClone) supplemented with 10% FBS. Cell lines were regularly tested negative for mycoplasma contamination using Mycoplasma PCR Detection Kit (Applied Biological Materials Inc; G238). Identity of the cell lines were verified by STR analysis.

Recombinant human TWEAK (PeproTech; 310-06) was reconstituted according to the manufacturer's recommendations. TNBC and HER2 cell lines were treated with 10 ng/ml TWEAK while ER-positive BC cell lines were treated with 30 ng/ml TWEAK.

For in vitro experiments, AP-1 inhibitor, T5224, (MedChemExpress; HY-12270) was dissolved in DMSO to a concentration of 10 mM and cells were treated with 10 μM T5224. For in vivo experiments, T5224 was dissolved in polyvinylpyrrolidone solution administered to mice orally at a concentration of 150 mg/kg.

## Clinical samples

Human TNBC tumour samples and their matched normal samples were collected from female patients who have not undergone neoadjuvant therapy nor had previous cancer treatment. Tissue samples were fresh frozen and stored in liquid nitrogen. Consent and approval for the use of clinical samples was obtained from the Nanyang Technological University Institutional Review Board, Singapore under the protocol Oncogenic TWEAK/Fn14 driven enhancer-promoter interactions in breast cancer (IRB: IRB-2021-388), in accordance with the Human Biomedical Research Act (HBRA) requirements. All clinical samples were obtained from SingHealth Tissue Repository Tissue Bank, Singapore.

## Plasmids

Fn14 CRISPR KO cells were generated using TLCV2 (Addgene; 87360) cloned with two different sgRNA guides (Supplementary Table. 1). NAMPT and mouse Fn14 CRISPR KD cells were generated using lenti-CRISPR v2 (Addgene; 52961) cloned with two different sgRNA guides (Supplementary Table. 1).

NAMPT enhancer deletions were generated by cloning the first sgRNA targeting upstream of the enhancer into TLCV2 and the second sgRNA targeting 1 kb downstream of the first sgRNA into pSpCas9(BB)-2A-GFP (PX458) (Addgene; 48138) which had Cas9 and EGFP replaced with RFP (Supplementary Table. 1).

shMAFG cells were generated using pLKO.1 (Addgene; 10878) cloned with shRNA sequences (Supplementary Table. 2).

The TWEAK overexpression construct was generated by amplifying the soluble TWEAK (sTWEAK) sequence from cDNA (Supplementary Table. 3) through PCR and this was cloned into pLJM1-EGFP (Addgene; 19319), replacing EGFP. Subsequently, pLJM1-sTWEAK was used as a template to amplify sTWEAK with a 3x Flag tag before being cloned into pLJM1-EGFP. In some experiments, hygromycin resistance was required, thus the hygromycin resistance sequence was amplified and cloned into pLJM1-Flag-sTWEAK by replacing the puromycin resistance.

pLenti CMV V5-LUC Blast (Addgene; 21474) was used for luciferase overexpression.

## CRISPR genotyping

*Fn14* KO MDA-MB-231 and Hs578T cells were identified through western blotting and then further verified through Sanger sequencing (Supplementary Table. 4).

*NAMPT* enhancer deletion MDA-MB-231 cells were identified through PCR and gel electrophoresis. Positive clones were further verified through Sanger sequencing (Supplementary Table. 4).

## Lentiviral production and transduction

For lentivirus production, $3 \times 10^6$ HEK293T cells were seeded per poly-l-lysine coated 10 cm plate. After 24 h, pMDL, VSVG, REV and the plasmid of interest was transfected into the cells via the calcium chloride method. 8 h later, the cells were washed with PBS and replenished with complete DMEM. After 24 h, the supernatant containing viral particles was harvested and filtered.

For lentiviral transduction, virus and 10 μg/ml polybrene was added to the cells. Media was changed 24 h after transduction and appropriate antibiotics was added 48 h after transduction.

## Mouse orthotopic xenograft

All animal experiments were performed in compliance with protocols approved by the Nanyang Technological University Institutional Animal Care and Use Committee (IACUC). Six- to ten-week-old female *Rag −/− IL2γ−/−* BALB/c and BALB/cN mice were housed at the animal facility of the Nanyang Technological University School of Biological Sciences.

When performing orthotopic xenograft, mice were anaesthetised with isofluorane and injected with cancer cells. In immunocompromised mice (*Rag−/− IL2γ−/−* BALB/c), $5 \times 10^5$ luciferase overexpressing MDA-MB-231 cells that also co-express combinations of sTWEAK overexpression, overexpression control, CRISPR *NAMPT* KD, CRISPR control, *NAMPT* enhancer #3 deletion and enhancer deletion control were injected. Immunocompromised mice were also injected with $1 \times 10^6$ MCF7 cells the harbour sTWEAK overexpression and the overexpression control. In regular BALB/cN mice, $3 \times 10^5$ luciferase overexpressing 4T1 cells that also co-express either lentiCRISPR *Fn14* KD or lentiCRISPR control were injected.

In AP-1 inhibition experiment utilising T5224, immunocompromised mice were injected with $5 \times 10^5$ luciferase and sTWEAK overexpressing MDA-MB-231 cells. Mice were treated with 150 mg/kg of T5224 or vehicle orally three times a week.

Primary tumour growth was measured weekly using callipers. For In vivo Imaging System (IVIS) imaging, 150 mg/kg D-luciferin (GoldBio) was injected intraperitoneally into the mice 10 min beforehand. The mice were then anaesthetised with isofluorane and imaged in an IVIS system (Perkin Elmer) under default settings. Images were analysed with Living Imaging Software (Perkin Elmer). Bioluminescent signals at metastatic sites were quantified by detecting the bioluminescent signal across the entire mouse and subtracting the bioluminescent signal from the primary tumour.

The mice were housed in temperatures of 21–25 °C, relative humidity (RH) of 55–60% and pressure (Pa) of 5–8 with a 12 h light-dark cycle.

## Western blotting

Cells were harvested and lysed for 30 min in RIPA buffer (50 mM Tris pH 8.0, 1 mM EDTA, 150 mM NaCl, 1% Triton X-100, 0.1% SDS, 0.1% sodium doxycholate) supplemented with cOmplete protease inhibitor cocktail (Roche; 5056489001), 5 mM sodium fluoride (Merck; S7920) and 1 mM DTT (Sigma-Aldrich; 10708984001). Whereas tissue samples were cut into small pieces, resuspended in RIPA buffer supplemented with protease inhibitor, sodium fluoride and DTT, then disrupted and homogenised with TissueRuptor II (Qiagen). The homogenate was incubated at 4 °C under mild agitation for 2 h, then centrifuged at 4 °C, max speed for 20 min and the supernatant was transferred to a new tube. For normal tissue samples with high fat content, they were further centrifuged and had their supernatant transferred to new tubes 2 additional times to ensure there was no carryover of fat.

The lysate concentration was then quantified using Bradford Assay dye solution (Bio-Rad; 5000006), lysate concentration from tissue samples were quantified using a BCA Protein Assay kit (Pierce; 23225). 30 μg of lysate was resolved on 8–12% Bis-Acrylamide gel (Bio-Rad; 1610158) and transferred to PVDF membrane (Bio-Rad; 1620177). The membranes were blocked using 5% skim milk in TBST buffer (20 mM Tris, 150 mM NaCl, 0.1% Tween 20) and incubated with the following primary antibodies overnight at 4 °C: anti-NFκB2 (Cell Signalling; 3017), anti-NF-κB p65 (Cell Signalling; 8242), anti-phospho-JNK (Cell Signalling; 4668), anti-JNK (Santa Cruz; 7345), anti-phospho-p44/p42 (ERK1/2) (Cell Signalling; 4370), anti-p44/p42 (ERK1/2) (Cell Signalling; 9107), anti-Fn14 (Abcam; 109365), anti-PBEF (NAMPT) (Santa Cruz; 393444), anti-MAFG (Santa Cruz; 166548) and anti-GAPDH (Santa Cruz; 32233). The next day, membranes were incubated for 1 h at room temperature with anti-rabbit secondary antibody (Cell Signalling; 7074) or anti-mouse secondary antibody (Santa Cruz; 516102). Antibodies used were diluted in 5% skim milk in TBST buffer at following dilutions: 1:10,000 for anti-GAPDH and secondary antibodies, and 1:1000 for all other antibodies. Protein bands were visualised using ChemiDoc MP Imaging System (Bio-Rad).

## Cell proliferation assay

$3 \times 10^4$ cells were seeded in complete media in a 12 well plate. Cells were incubated overnight to allow attachment. The cells were then starved for 8 h in serum-free DMEM before being treated with or without TWEAK. Cells were counted at 0 h, 24 h and 48 h after addition of TWEAK. The number of cells counted were normalised against the 0 h cell count. For AP-1 inhibition experiments, the assay was repeated with the addition of 10 μM T5224 or vehicle. The experiment was performed in 3 biological replicates.

## Invasion assay

Invasion assay was performed with 24 well Transwell inserts (Corning) that were coated with 200 μg/ml Matrigel (Corning). BC cell lines were starved in optiMEM (Gibco) for 8 h, followed by treatment with or without 10 ng/ml TWEAK. After 24 h, cells were trypsinised, counted and resuspended in optiMEM to a concentration of $2.5 \times 10^5$ cells/ml. TWEAK was also added to the samples previously treated with TWEAK. Before seeding the cells, excess liquid was removed from the upper chamber of the transwell inserts and the inserts are placed into a 24 well plate containing 600 μl complete DMEM. 100 μl of the cell suspension was then added to the upper chamber of the transwell insert. The cells were then incubated at 37 °C for 24 h. After which, the transwell inserts were fixed in 70% ethanol for 10 min, left to dry for 10 min and stained in 0.2% crystal violet for 10 min. Non-invading cells in the upper chamber were then removed with cotton buds. Invaded cells were visualised under an inverted microscope and images were taken from four random frames per insert. The number of invaded cells per frame was counted and averaged. For AP-1 inhibition experiments, the assay was repeated with the addition of 10 μM T5224 or vehicle. The experiment was performed in 3 biological replicates.

## NAD/NADH and ATP metabolic assays

Cells were seeded in complete media in a 6 well plate and incubated overnight to allow attachment. The next day, cells were starved for 8 h in serum-free DMEM before being treated with or without TWEAK for 48 h. After treatment, cells were trypsinised, counted and resuspended in PBS at a concentration of $8 \times 10^4$ cells/ml. TNBC tumours used were homogenised in PBS at a concentration of 20 μl/gram. NAD/NADH and ATP assays were performed using NAD/NADH-Glo assay kit (Promega; G9071) and CellTiter-Glo 2.0 assay kit (Promega; G9241), respectively. Assays were performed in 384 well plates according to the manufacturer's protocol. The luminescence recorded for each sample was averaged, had the background subtracted. For samples from cell lines, values were normalised against the untreated control whereas in tissue samples, tumour values were normalised against their matched normal samples.

## Filopodia staining and microscopy

$5 \times 10^4$ cells/treatment condition were seeded and grown on a coverslip in a 6 well dish overnight for attachment. On the next day, cells

were starved for 8 h in serum-free DMEM followed by treatment with or without TWEAK for 48 h. Following treatment, medium was removed, and cells were fixed with 3.7% formaldehyde (Sigma) in 1X PBS for 20 min at room temperature. Once fixed, cells were washed 3 times with 1X PBS and incubated in permeabilization buffer containing 0.2% Triton X-100 in PBS for 15 min. After permeabilization, cells were washed 3 times with 1X PBS and incubated with Alexa Fluor™ 488 Phalloidin (diluted 1.5:200 in PBS from 400X stock solution in DMSO) (Invitrogen; 12,379) for 30 min in the dark. Following staining, cells were washed 3 times in 1X PBS. Finally, coverslips were mounted onto a microscopic glass slide using ProLong Gold antifade with DAPI reagent (Invitrogen; P36914) and viewed under fluorescence microscope using a 63x objective and a FITC filter. Images were further analysed using Fiji software[87] and filopodia density (Number of filopodia/edge length) was calculated per image for each treatment condition using FiloQuant[88]. Total of three frames were taken for analysis.

### RNA library construction, sequencing and analysis
Total RNA was isolated using TRIzol Reagent (Invitrogen) according to the manufacturer's protocol and Dnase-treated with TURBO DNA-free kit (Invitrogen). The RNA quality was verified using Agilent RNA 6000 kit (Agilent Technologies; 5067-1511) and Agilent Bioanalyzer System (Agilent Technologies). Next, poly-A mRNA was isolated using NEB-Next Poly(A) mRNA Magnetic Isolation Module (NEB; E7490) and subsequently used to assemble the RNA-seq library using NEBNext® Ultra™ II Directional RNA Library Prep Kit for Illumina (NEB; E7760), according to the manufacturer's protocol. RNA-seq libraries were sequenced on HiSeqX. All RNA-seq experiments were performed with three biological replicates.

The quality of sequencing reads was assessed with FastQC[89]. The library adaptors were removed using Trimmomatic[90]. RNA-seq reads were then mapped to hg38 with STAR[91] and the raw counts were extracted using featureCounts[92]. Differential analysis was performed with DESeq2[93] where differentially expressed genes were defined by a FDR ≤ 0.1. Gene ontology analysis was performed using PANTHER[94,95]. Pathway analyses were performed using the GSEA software[96–98]. Heatmaps were generated using R Statistical Programming Language and ggplot2[99].

### Integration with TCGA BRCA RNA-seq dataset
TCGA BRCA RNA-seq sample counts were retrieved using the UCSC Cancer Browser (http://xena.ucsc.edu/welcome-to-ucsc-xena/)[100].

### scRNA-seq analysis
scRNA-seq dataset was acquired from[42] where patient 5 (TNBC) primary tumour and both lymph node (metastasis-positive) samples were used for analysis. scRNA-seq analysis was performed using Seurat[101] as described by the authors. Cell type classification was performed using SingleR[102]. Plots depicting average *Fn14* and *TWEAK* expression were generating using R Statistical Programming Language.

### ChIP qPCR, sequencing library construction, sequencing and analysis
Cells and clinical samples were crosslinked with 1% Formaldehyde (Sigma) for 10 min and quenched with 0.125 M glycine for 5 min at room temperature. Cells were washed with cold PBS and lysed with a SDS lysis buffer supplemented with protease inhibitor (Roche). Crosslinked clinical samples were washed thrice with cold PBS, disrupted and homogenised with TissueRuptor II (Qiagen) before being lysed with a SDS lysis buffer supplemented with protease inhibitor. Lysed samples were then sonicated with Bioruptor Plus (Diagenode) to achieve DNA fragments ranging between 200 and 500 bp. Chromatin was precleared with blocked Protein G Sepharose beads (GE Healthcare) for 2 h at 4 °C before being rotated overnight at 4 °C with Anti-acetyl-Histone H3 (Lys27) (Sigma; 07-360) or Anti-MAFG (GeneTex;

GTX114541) antibody. Following that, blocked Protein G Sepharose beads were added and rotated at 4 °C for 6 h. The beads were subsequently washed in the following order of buffers: SDS buffer, high salt buffer, lithium chloride wash buffer and Tris-EDTA buffer. Beads were then eluted in ChIP elution buffer overnight at 65 °C. Eluted ChIP DNA was purified using QIAquick PCR purification kit (Qiagen).

Eluted ChIP DNA was used for ChIP qPCR and ChIP-seq library preparation. ChIP qPCR was performed using SsoAdvanced Universal SYBR mix (Bio-rad; 1725272). Ct values were normalised to relative amount of 1% of input DNA. The ChIP qPCR primer sequences used can be found in Supplementary Table 5. ChIP-seq libraries were prepared using NEBNext Ultra II DNA Library Prep Kit (NEB; E7645) according to the manufacturer's protocol. Size and quality of libraries were verified using the Agilent high sensitivity DNA kit (Agilent Technologies; 5067-4626) and Agilent Bioanalyzer System (Agilent Technologies). Libraries were then quantified using the KAPA library quantification kit (Roche; KK4824) and pooled. ChIP DNA libraries were sequenced on HiSeqX. All cell line-based ChIP-seq experiments were performed with at least two biological replicates.

ChIP-seq reads were assessed and the adaptors were removed as performed with the RNA-seq data. Reads were then mapped to hg38 using Bowtie2[103] and peaks were called using MACS2 with subtraction of input signal. Blacklisted regions were then removed from the called peaks using bedtools[104]. Differential analysis between ChIP peaks was performed with DiffBind[105]. Heatmaps showing signal distribution over ChIP peak regions were generated using deepTools[106]. Prediction of biological functions associated with differential enhancers was performed with GREAT[107]. Plots of ChIP-seq peaks were produced with pyGenomeTracks[108,109].

SE calling was performed using the ROSE algorithm[110,111]. Differentially regulated SEs were identified by overlapping SEs with differentially regulated enhancers. Common SEs between cell lines were defined as SEs that had at least 5 kb overlap between them.

### ATAC-seq library construction, sequencing and analysis
ATAC-seq was performed as described in ref. 112 with some modifications. In brief, $5 \times 10^4$ cells were lysed in ATAC Resuspension buffer containing NP40, TWEEN-20 and Digitonin. While clinical samples were disrupted and homogenised with TissueRuptor II (Qiagen) and nuclei were isolated through Iodixanol density gradient centrifugation. Subsequently, lysed cells or nuclei were transposed in a transposition mix containing digitonin and Tn5 transposase for 30 min at 37 °C while shaking at 1000 rpm. The transposition reaction was then cleaned up with DNA Clean and Concentrator kit (Zymo), followed by amplification through PCR with Nextera DNA CD Indexes (Illumina) and NEB-Next Ultra II Q5 Master mix (NEB). Amplified ATAC-seq libraries were then size selected with AMPure XP Reagent (Beckman Coulter) and sequenced on HiSeqX. All cell line based ATAC-seq experiments were performed with at least two biological replicates.

ATAC-seq reads were assessed, library adaptors removed and then mapped to hg38 as performed with the ChIP-seq data. Following which, mitochondrial reads were filtered with Bamtools[113], duplicate reads were removed with Picard and peaks were called with MACS2. Differential analysis between ATAC-seq peaks was performed using DiffBind and the heatmaps were plotted using deepTools. TF Footprinting and differential TF binding analysis was performed using TOBIAS[114] where volcano plots were generated using R.

### HiChIP sequencing and analysis
$2 \times 10^7$ MDA-MB-231 cells were treated with 10 ng/ml TWEAK for 48 h. The cells were pelleted, stored at −80 °C and sent to Dovetail Genomics (part of Cantata Bio, LLC.) for acetyl-H3K27 HiChIP library preparation and sequencing.

Sequencing reads were aligned to hg38 with BWA-MEM[115]. Next, valid ligation events were recorded and sorted, PCR duplicates were

then removed and the bam files were generated with pairtools and SAMtools[116]. HiChIP loops were then called using FitHiChIP[117] and differential analysis was performed with diffloop[118].

## Statistics and reproducibility

Two-way ANOVA statistical test was used in all in vitro assays while *t* test was used to test for statistical significance in vivo tumour growth and metastasis assays. All in vitro assays and western blots were performed in 3 biological replicates while all in vivo assays were performed in 5 biological replicates. Experiments involving cell lines were performed 3 times independently, each time on different days. Asterisks represent the degree of statistical significance, $*P < 0.05$; $**P < 0.01$; $***P < 0.001$. All statistical analyses and graphics were performed using R Statistical Programming Language or GraphPad Prism software.

No statistical method was used to predetermine sample size. No data were excluded from the analyses. No experiments were randomised. The investigators were not blinded to allocation during experiments and outcome assessment.

## Data availability

The ChIP-seq, RNA-seq, ATAC-seq and HiChIP data generated in this study have been deposited in the GEO (Gene Expression Omnibus) database under accession code GSE231483. Source data are provided with this paper.

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

## Acknowledgements

This study is funded by the National Research Foundation (NRF) Singapore, under its Singapore NRF Fellowship (NRF-NRFF2018-04). In addition, we thank the Nanyang Assistant Professorship (NAP) Start-up-grant to Y.L.'s lab and Nanyang Technological University for the PhD scholarship funding of N.S.

## Author contributions

Y.L. conceptualised the ideas to this manuscript and supervised the study. Y.L. and N.S. planned and devised the experiments. N.S. performed and analysed all molecular and cell biology experiments, with assistance from K.D. in western blotting, cell sorting, genotyping and microscopy. B.K.T.T. and Y.S. contributed to the TNBC patient analysis

and tissue samples. N.S. performed the tumour xenograft studies, while S.M.T. contributed the *Rag–/– IL2γ–/–* BALB/c model and technical expertise. N.S. performed the computational analyses and data visualisation, with input from J.M.C. in the RNA-seq, ATAC-seq and HiChIP analyses. N.S. and Y.L. co-wrote the manuscript, with input from J.M.C., K.D., and Y.S.

## Competing interests

The authors declare no competing interests.
