## [Peer Review File · Nature Communications]

TWEAK/Fn14 signalling driven super-enhancer reprogramming promotes pro-metastatic metabolic rewiring in Triple-Negative Breast CancerREVIEWER COMMENTS

Reviewer #1 (Remarks to the Author):

In this work Sin and colleagues dissect the signalling pathway TWEAK/Fn14 in breast cancer via an extensive series of experiments. They begin by observing that Fn14 (which I believe should be listed with its official name TNFRSF12A) is relatively more expressed in basal TNBC tumours (ER negative) as compared to other subtypes. They then leverage cell lines (ER+ and ER-) to show that activating the pathway via TWEAK exposure leads to large-scale chromatin event including changes in chromatin accessibility, H3K27ac and looping which all correlate nicely with the transcriptional changes observed in their models. These changes are often recapitulated (at least partially) in patient-derived samples. They converge on a specific enhancer which regulates the NAMPT gene and link these metabolic changes to filopodia formation concluding that TWEAK contributes to increase invasion via metabolic reprogramming. I enjoyed reading the manuscript and I think it contains some interesting observations. There are of course some points which should be debated/clarified in a revised submission.

1- I could not understand, but maybe I missed this, what cell type would provide the ligand in physiological conditions. The authors use TWEAK stimulation in their in vitro work, where would these stimuli come from in primary and metastatic settings? Do the authors have any evidence that physiological stimuli would lead to the same level of activation of the pathway as observed in vitro? What would drive the "constitutive activation" which in their words contribute to TNBC poorer outcome (which I thought was due to the lack of targeted therapy, not some inherent biological difference, at least this is what their intro is suggesting)

2- The comparison tumour normal are somewhat inappropriate as the tumour is largely composed by clonal cells expanding from the initial transformed cell (cell of origin). The normal tissues here are presumably composed of a collection of cell types (ductal cell, ER+ and ER-, fibroblasts, adipocytes, immune etc...). The authors should provide some IHC staining of their sample to put the RNA into context.

3- I am somewhat surprised by the strong enrichment of FN14/ TNFRSF12A in ER- cells. A quick look at the protein atlas seems to suggest that this receptor is strongly overexpressed in normal ER+ cells (which again presumably are not the cell of origin for these tumours).

4- The authors use the term Epigenetic switch/plasticity throughout the manuscript. It is not clear if they argue that TWEAK/Fn14 remodel the epigenome of cancer cells to give them a fitness advantage (invasion) which will ultimately contribute to metastatic formation or if this is just a transitory effect due to the presence of the stimuli. I think the authors should at least perform some ATAC/CHIP-seq after the stimuli has been washed off to get a sense of what they are observing. I would say that epigenetic reprogramming is a more stable event while here it is very hard to judge if what they have characterized is a classical signalling response pathway (which of course would involve K27ac and chromatin changes at target genes).

5- I am struggling with the intro of the manuscript and how that is relevant to the results. The manuscript has nice data on the activity of the pathway and really dig down into the effector of the phenotype. I cannot comment on how novel/unexpected these results are, but the quality of the data is there. However, they introduce this work but stating that TNBC lack targeted therapy (by the way, they forgot to mention that I/O is now standard of care in Neoadjuvant regimens) and this is why TNBC has more relapse (which is also incorrect, as late relapse in ER+ means that overall the number of death and relapse is much higher for luminal patients overall). How does this epigenetic study relate to better therapeutic targets? Are the authors proposing some sort of metabolic intervention to prevent metastatic outgrowth? Or even an enhancer targeting strategy? It would be worth to remember that epigenetic therapies target the entire epigenome, not a single enhancer. In the absence of some data on the translatability of these findings, I would suggest that the authors remodel the intro to better reflect the actual findings.

6- Can the authors clarify/add details in the text how the epigenetic analysis on real tumour samples was done? When they show WB, it seems there were few patients involved. When they show ATAC/K27ac it's not clear if they combined material into one experiment or is just one sample/patient.

7- Finally, is the lack of changes in ER+ cells due to lack of the receptor or different biology? It seems that MCF7 and T47D have a little bit of protein (while CAMA1 are negative, which then easily explain the lack of response). However, in the epigenetic analysis heatmaps the authors only

shows differential peaks in ER- cells and a lack of changes in the same loci for ER+. This of course does not rule out that TWEAK is also dramatically changing the chromatin landscape of MCF7 cells. A genome wide profile for the ER+ cells would be a nice figure to add to the supplementary (or maybe I missed it). And once more, for these heatmaps, comparing normal vs tumor to cell lines with or without TWEAK is not appropriate as they do not ask the same question. The cell line show activity of enhancer in response to a well-defined stimuli. The normal vs TNBC shows, well, I am not sure what it shows.... the result could be driven by clonal expansion, heterogeneity of one tissue vs the other, but surely do not represent TWEAK exposure in vivo.

8- The enhancer 3 results are impressive, and I'd like to congratulate the authors. They should take center stage of this work, especially the dissection of what regulates that enhancer and how this could be exploited possibly in the clinical setting.

Reviewer #2 (Remarks to the Author):

The current manuscript from Nicholas et al. describes epigenetic mechanisms contributing to TWEAK/Fn14 signaling in triple-negative breast cancer. They used several breast cancer models in combination with publicly available datasets (TCGA) to determine whether TWEAK/Fn14 signaling was more relevant to TNBC compared to luminal breast cancer. While TWEAK/fn14 signaling has been investigated in mammary gland morphogenesis and Her2+ breast cancer progression and invasion, this manuscript emphasizes on the transcriptional regulatory role of super-enhancers (SE). The study reveals an association between TWEAK/fn14 signaling and metabolic alteration. A major conclusion is that TWEAK/fn14 signaling could activate NAMPT expression via a SE-Promoter crosstalk leading to increased NAD+ production. Overall, the study is interesting but needs to be improved.

General comments:

1- A major limitation of this investigation relates to the origin of TWEAK which remains elusive from a biological perspective. Considering the complexity of the tumor microenvironment, and previous implication of TWEAK/fn14 signaling stromal cell differentiation processes (e.g., endothelial cells, fibroblasts), a thorough evaluation of TWEAK/fn14 expression in bulk tumors is necessary. Perhaps both TWEAK and fn14 are expressed from the same cells which would suggest an autocrine mechanism.

2- Similarly to TNBCs, Fn14 is also highly expressed in Her2 breast cancer subtypes (Fig.1a). Therefore, it would make sense to evaluate whether the identified mechanisms remain relevant for Her2+ breast cancer. Additionally, how to uncouple the TWEAK/fn14-mediated epigenetic changes from genetic alterations remains a concern.

Specific comments:

Major comments:

1- Fig.2d suggests that similar pathways could be activated in both TNBC and luminal BC. However, the TWEAK/fn14 signaling associates with poor survival outcomes in TNBC only. Therefore, one would wonder if such differences were driven by TNBC-specific signaling pathways as indicated in Fig.2d (e.g., NAD Biosynthesis, MYC, methionine, JNK, cJun). Interestingly, these results also indicate a TWEAK-mediated pathways enrichment in Luminal models. How do you reconcile these cell-specific pathways to tumor progression and overall survival outcomes in breast cancer patients?

2- To assess TWEAK/fn14 signaling, most epigenetic experiments have been based on TWEAK exposure. As fn14 knockout models have been established and characterized (Sup.Fig1), including more genetic approaches to strengthen the epigenetic and transcriptional role of fn14 will significantly strengthen the results. For instance, assessing epigenetic changes using fn14 KO cells +/- TWEAK would be indicated.

3- The authors claim that NAMPT is specifically induced in TNBC but not in Luminal models (Fig6a,b, and e). For such claims to be true based on western blot experiments, a similar amount of protein must be loaded for each sample. It also remains possible that the lack of expression in luminal cells is related to genomic differences (e.g., mutations). This needs further evaluation. Contrasting with the TNBC models, CAMA1 shows a decreased level of NAMPT following TWEAK treatment. How do the authors interpret this result?

4- The ChiP-PCR results suggest that MAFG mediates TWEAK/fn14 signaling at NAMPT enhancer #3. However, it remains unclear why only enhancer #3 is impacting NAMPT transcription and no other enhancers. It seems adequate to also evaluate the impact of MAFG on Enhancers #1 and #2. Moreover, assuming that NAMPT upregulation is specific to TNBC, the expression and binding of MAFG should be evaluated in other breast cancer subtypes. This experiment is determinant considering that TWEAK treatment appears to increase fn14 level in luminal breast cancer as well (see Fig.2e).

5- The overexpression of WEAK in MCF7 cells shows a decreased trend in primary tumor proliferation which may indicate an opposite function of TWEAK/fn14 signaling in luminal breast cancer models when compared to TNBC. Unfortunately, only 3 mice were included in each group (Sup. Fig.2b-c). Including more mice would help strengthen the conclusions.

6- In Fig.6j, it is unclear how the metastatic sites were defined and quantified. Is this based on ex vivo imaging? Additionally, including representative metastatic lesions is required to ascertain the origin of bioluminescence signals. Assessing differences at the primary site may also be relevant.

7- Fig.6d and Fig.6e: A more quantitative approach with statistics would be beneficial.

8- The enhanced AP-1 binding activity in TNBC following TWEAK exposure suggests that AP-1 mediates TWEAK/fn14 signaling in these models. These results require further functional validation to determine whether the impact of TWEAK/fn14 on cell proliferation, migration, invasion, and overall metastasis in vivo is truly mediated by AP1 signaling. Additionally, it is unclear how MAFG is recruited at enhancer sites following TWEAK/fn14 activation. Is it known whether NFKB could mediate this process? Also, assessing the association (correlation) between MAFG and NAMPT from patient datasets would strengthen the clinical relevance of the mechanism.

Minor Comments:

9- In Sup. Fig.3, the survival follow-up stops at month 60 while other breast cancer subtypes reach month 120. Is there any particular reason for such variation?

Reviewer #3 (Remarks to the Author):

In the manuscript "TWEAK/Fn14 signalling driven super-enhancer reprogramming promotes pro-metastatic metabolic rewiring in Triple-Negative Breast Cancer" the authors propose that expression of Fn14 is a potential driver of epigenetic and metabolic reprogramming in Triple Negative Breast Cancer (TNBC). This is an interesting manuscript and may have potential clinical significance. However, there are several significant pitfalls that need to be resolved before the conclusions of the manuscript can be supported by the data.

Major: The main issue with the manuscript is the paucity of clinical data sets. The manuscript is heavily dependent on the use of breast cancer cell lines in vitro. The use of avatars and also of immune competent mouse models is imperative to confirm the conclusions.

Cell lines may not reflect the clinical cancer and its microenvironment. Therefore, at this point the conclusions are based on cultured cells.

It is also important to highlight that culture conditions of these cells are often very artificial and may

significantly change the outcome of metabolism and its epigenetic conditions.
Thus, the validation of the data is clinically relevant data sets is imperative.

Revisions and Responses to Reviewers' comments

Reviewer A Comments

Reviewer #1 (Remarks to the Author):

In this work Sin and colleagues dissect the signalling pathway TWEAK/Fn14 in breast cancer via an extensive series of experiments. They begin by observing that Fn14 (which I believe should be listed with its official name TNFRSF12A) is relatively more expressed in basal TNBC tumours (ER negative) as compared to other subtypes. They then leverage cell lines (ER+ and ER-) to show that activating the pathway via TWEAK exposure leads to large-scale chromatin event including changes in chromatin accessibility, H3K27ac and looping which all correlate nicely with the transcriptional changes observed in their models. These changes are often recapitulated (at least partially) in patient-derived samples. They converge on a specific enhancer which regulates the NAMPT gene and link these metabolic changes to filopodia formation concluding that TWEAK contributes to increase invasion via metabolic reprogramming. I enjoyed reading the manuscript and I think it contains some interesting observations. There are of course some points which should be debated/clarified in a revised submission.

Comment A1

I could not understand, but maybe I missed this, what cell type would provide the ligand in physiological conditions. The authors use TWEAK stimulation in their in vitro work, where would these stimuli come from in primary and metastatic settings? Do the authors have any evidence that physiological stimuli would lead to the same level of activation of the pathway as observed in vitro? What would drive the "constitutive activation" which in their words contribute to TNBC poorer outcome (which I thought was due to the lack of targeted therapy, not some inherent biological difference, at least this is what their intro is suggesting)

Response A1

To identify the cell types that provide TWEAK ligand in physiological conditions, we analysed scRNA-seq data from a TNBC primary tumour and its matched lymph node samples where metastasis was detected, using data from Xu *et al.* (1). Here, we found that TWEAK is mainly expressed in endothelial cells in the primary tumour setting; whereas in the metastatic setting, TWEAK is mainly expressed in monocytes and macrophages (Fig. 1 and 2 (Extended Data Fig. 1e,f in the manuscript)). Our analysis suggests that these cell types are the main contributors of TWEAK stimuli in their respective settings. We have included this data in lines 94-100 of the revised manuscript.

The TWEAK (TNFSF12) cytokine has been reported to activate the non-canonical NF- κ B pathway at low, physiological concentrations (2, 3). This has been verified in our TWEAK induced cell lines (Fig. 2e in the manuscript). Additionally, the tumour-promoting effects of stimulation with a physiological dose of TWEAK have also been further described in glioma cells (4).

To clarify, constitutive activation of TWEAK/Fn14 signalling would be driven by persistent engagement of Fn14 by the TWEAK cytokine to promote TWEAK/Fn14 activation. As TNBC patients and cell lines exhibit enhanced Fn14 levels, we suggest that this leads to increased TWEAK/Fn14 activation and consequent cellular processes which contribute to poorer TNBC outcome.

Figure 1. Plot depicting the average *Fn14* expression of single cells in the primary TNBC tumour and its matched lymph node metastases. T-test was used for statistical analysis comparing epithelial cells versus all other cell types. * $P < 0.05$; ** $P < 0.01$; *** $P < 0.001$.

Figure 2. Plot depicting the average *TWEAK* expression of single cells in the primary TNBC tumour and its matched lymph node metastases. T-test was used for statistical analysis comparing endothelial cells versus all other cell types, monocytes versus all other cell types except macrophages and macrophages versus all other cell types except monocytes. * $P < 0.05$; ** $P < 0.01$; *** $P < 0.001$.

Comment A2

The comparison tumour normal are somewhat inappropriate as the tumour is largely composed by clonal cells expanding from the initial transformed cell (cell of origin). The normal tissues here are presumably composed of a collection of cell types (ductal cell, ER+ and ER-, fibroblasts, adipocytes, immune etc...). The authors should provide some IHC staining of their sample to put the RNA into context.

Response A2

The comparison between TNBC tumour sample and their matched normal sample demonstrates that in TNBC samples, Fn14 expression is elevated in most tumour tissues relative to adjacent normal breast tissues in the same patient. To place the TWEAK/Fn14-driven alterations identified in TNBC cell lines in the context of TNBC tumour-specific signatures, we further compared these *cis*-regulatory changes with the enhancer landscape of TNBC patient samples. This illustrates that following TNBC development in patients, the epigenetic changes observed in cell lines are also detected in patients, implying that they could be important for TNBC tumour growth and metastasis.

We have also leveraged on existing patient datasets which show similar results that are consistent with our findings to put the RNA into context. A previous report has demonstrated through IHC staining that ER-negative breast cancer tumours express higher levels of Fn14 compared to ER-positive breast cancer tumours and normal breast tissue (5). To confirm this, we interrogated the TCGA BRCA RNA-seq dataset, consisting of ~1200 patients, which also indicated that Fn14 levels are significantly elevated in ER-negative breast cancer tumours compared to ER-positive breast cancers and normal breast tissue (Fig. 1a in manuscript).

Comment A3

I am somewhat surprised by the strong enrichment of FN14/ TNFRSF12A in ER- cells. A quick look at the protein atlas seem to suggest that this receptor is strongly overexpressed in normal ER+ cells (which again presumably are not the cell of origin for these tumours). See figure below.

Response A3

Thank you for the comment. It is important to note that the single cell Human Protein Atlas data is based on healthy tissue data. Therefore, in a healthy breast tissue setting, it is possible that healthy ER+ cells express higher Fn14 levels. However, upon cancer development, Fn14 levels can be dysregulated in tumour tissues, as reported in multiple cancers including breast cancer (6-8). Here in our study, we observed that Fn14 expression is elevated in TNBC samples compared to ER-positive breast cancer and normal breast tissue samples, as shown in the TCGA BRCA RNA-seq dataset (Fig. 3 (Fig. 1a of the manuscript)) and our matched patient sample data (Fig. 4 (Fig. 1b of the manuscript)).

Figure 3. Relative *Fn14* gene expression levels in Basal-like, HER2, Luminal A, Luminal B, Normal-like and non-cancer patient samples from the TCGA BRCA RNA-seq dataset.

Figure 4. Fn14 protein expression analysed by western blotting in (b) TNBC patient tumours (T) and their matched normal samples (N).

Comment A4

The authors use the term Epigenetic switch/plasticity throughout the manuscript. It is not clear if they argue that TWEAK/Fn14 remodel the epigenome of cancer cells to give them a fitness advantage (invasion) which will ultimately contribute to metastatic formation or if this is just a transitory effect due to the presence of the stimuli. I think the authors should at least perform some ATAC/CHIP-seq after the stimuli has been washed off to get a sense of what they are observing. I would say that epigenetic reprogramming is a more stable event while here it is very hard to judge if what they have characterized is a classical signalling response pathway (which of course would involve K27ac and chromatin changes at target genes).

Response A4

Thank you for raising this issue. To verify this, we treated the TNBC cell lines, MDA-MB-231 and Hs578T, with TWEAK for 48h followed by a 96h wash-off period. In doing so, we found that TWEAK wash-off reverted TWEAK/Fn14-driven non-canonical NF- κ B and JNK activation (Figure 5a (Supplementary Fig. 3a in manuscript)). In addition, H3K27ac ChIP-seq also revealed that the TWEAK-induced enhancer changes are reverted to levels similar to untreated controls following TWEAK wash-off (Figure 5b (Supplementary Fig. 3b in manuscript)). Together, this indicates that TWEAK/Fn14 activation is a transitory effect, but when persistently activated in a tumour setting (by microenvironmental stimuli), it leads to alterations in biochemical signalling and the epigenetic landscape that promote TNBC invasion and metastasis.

Figure 5. (a) NF- κ B and JNK pathway signalling regulators analysed by western blotting in untreated, TWEAK-treated and TWEAK-treated with 96h wash-off MDA-MB-231 and Hs578T cells. (b) H3K27ac ChIP-seq signals of untreated, TWEAK-treated and TWEAK-treated with 96h wash-off MDA-MB-231 and Hs578T cells at TWEAK/Fn14-driven differential TNBC enhancer sites.

Comment A5

I am struggling with the intro of the manuscript and how that is relevant to the results. The manuscript has nice data on the activity of the pathway and really dig down into the effector of the phenotype. I cannot comment on how novel/unexpected these results are, but the quality of the data is there. However, they introduce this work but stating that TNBC lack targeted therapy (by the way, they forgot to mention that I/O is now standard of care in Neoadjuvant regimens) and this is why TNBC has more relapse (which is also incorrect, as late relapse in ER+ means that overall the number of death and relapse is much higher for luminal patient overall). How this epigenetic study relates to better therapeutic targets? Are the authors proposing some sort of metabolic intervention to prevent metastatic outgrowth? Or even an enhancer targeting strategy? It would be worth to remember that epigenetic therapies target the entire epigenome, not a single enhancer. In the absence of some data on the translatability of these findings, I would suggest that the authors remodel the intro to better reflect the actual findings.

Response A5

Thank you for highlighting this. We have revised the introduction from lines 33 to 42 to mention I/O and also clarified that despite the existing therapies, few patients achieve pathologic complete

response. As such, there is an urgent need for novel targeted therapies that could improve TNBC patient outcome. Our study describes the overexpression of Fn14 in TNBCs which can drive distinct signalling pathways, transcriptomic and epigenetic alterations, as well as metabolic effects which ultimately contribute to TNBC metastasis. These findings highlight the therapeutic potential of targeting the TWEAK/Fn14 signalling cascade and future characterisation of the oncogenic functions of its downstream gene targets for plausible intervention. As suggested, we have revised the introduction to reflect the significance of our findings.

Comment A6

Can the authors clarify/add details in the text how the epigenetic analysis on real tumour samples was done? When they show WB, it seems there were few patients involved. When they show ATAC/K27ac it's not clear if they combined material into one experiment or is just one sample/patient.

Response A6

We understand the confusion and would like to clarify this. The ATAC-seq and H3K27ac ChIP-seq analyses on the TNBC and matched normal tissue samples only involved paired tissues where the tumour Fn14 expression was higher than its matched normal sample. Specifically, we used the following samples for epigenetic analyses: 2086, 8370, 2963, 6122, 8850 and 5929, where we merged the ATAC-seq or ChIP-seq signals of all 6 TNBC tumours and their matched normal samples. We have also revised the figure legends, in Figures 3a, 3d, 4a, 6c and Extended Data Figure 5 of the manuscript to reflect this.

Comment A7

Finally, is the lack of changes in ER+ cells due to lack of the receptor or different biology? It seems that MCF7 and T47D have a little bit of protein (while CAMA1 are negative, which then easily explain the lack of response). However, in the epigenetic analysis heatmaps the author only shows differential peaks in ER- cells and a lack of changes in the same loci for ER+. This of course does not rule out that TWEAK is also dramatically changing the chromatin landscape of MCF7 cells. A genome wide profile for the ER+ cells would be a nice figure to add to the supplementary (or maybe I missed it). And once more, for these heatmaps, comparing normal vs tumor to cell lines with or without TWEAK is not appropriate as they do not ask the same question. The cell line show activity of enhancer in response to a well-defined stimuli. The normal vs TNBC shows, well, I am not sure what it shows.... the result could be driven by clonal expansion, heterogeneity of one tissue vs the other, but surely do not represent TWEAK exposure in vivo.

Response A7

Epigenetic analyses of the changes in ER-positive breast cancer cells can be found in Figures 6,7 (Extended Data Fig 5a,c in the manuscript). Here, we demonstrate that TWEAK/Fn14 signalling can alter the chromatin accessibility and enhancer landscape of ER-positive breast cancer cells as well. However, we note that these changes are not subtype specific as these regions were also regulated in a similar manner in the TNBC cell lines.

The comparison of normal vs tumour to cell lines treated with or without TWEAK demonstrates that the TWEAK/Fn14-driven epigenetic changes observed in cell lines are also present in patients upon TNBC development and Fn14 overexpression. In doing so, this highlights the potential for these pro-oncogenic epigenetic alterations to drive tumour growth and metastasis in patients.

Figure 6. ATAC-seq signals of MDA-MB-231, Hs578T, SKBR3 and MCF7 with and without TWEAK, as well as the merged signal of Fn14 high TNBC patient tumours and their matched normal samples (2086, 8370, 2963, 6122, 8850 and 5929) at differentially regulated chromatin accessible sites in MCF7.

Figure 7. H3K27ac ChIP-seq signals of MDA-MB-231, Hs578T, SKBR3, MDA-MB-453, MCF7 and CAMA1 with and without TWEAK, as well as the merged signal of Fn14 high TNBC patient tumours and their matched normal samples at differentially regulated enhancer sites in the ER-positive cell lines.

Comment A8

The enhancer 3 results are impressive, and I'd like to congratulate the authors. They should take center stage of this work, especially the dissection of what regulates that enhancer and how this could be exploited possibly in the clinical setting.

Response A8

We thank Reviewer #1 for the kind comments. To further verify the role of *NAMPT* enhancer #3, we have performed additional MAFG ChIP-seq experiments and demonstrated that MAFG binds only to *NAMPT* enhancer #3 in TNBC cell lines (Figure 8 (Extended Data Fig. 7f in manuscript)). Consistent with this, knockdown of MAFG inhibits TWEAK/Fn14-driven activation of enhancer #3 but not in enhancers #1 and #2 (Figure 9 (Extended Data Fig. 8 in manuscript)).

Figure 8. *NAMPT* locus. Top: MAFG ChIP-seq of untreated and TWEAK treated MDA-MB-231, Hs578T, SKBR3 and MCF7 cells. Bottom: H3K27ac ChIP-seq of untreated and TWEAK-treated MDA-MB-231 cells.

Figure 9. (a) MAFG and NAMPT protein expression analysed by western blotting in Control and shMAFG MDA-MB-231 and Hs578T cells treated with and without TWEAK. Plots depicting the IgG and H3K27ac ChIP qPCR %input at (b) *NAMPT* enhancer #1, (c) *NAMPT* enhancer #2 and (d) *NAMPT* enhancer #3 in control and shMAFG MDA-MB-231 and Hs578T cells treated with and without TWEAK (mean \pm s.d). Data shown represent n=3 biological replicates. Two way ANOVA was used for statistical analysis where *P < 0.05; **P < 0.01; ***P < 0.001.

Reviewer B Comments

REVIEWER#2

The current manuscript from Nicholas et al. describes epigenetic mechanisms contributing to TWEAK/Fn14 signaling in triple-negative breast cancer. They used several breast cancer models in combination with publicly available datasets (TCGA) to determine whether TWEAK/Fn14 signaling was more relevant to TNBC compared to luminal breast cancer. While TWEAK/fn14 signaling has been investigated in mammary gland morphogenesis and Her2+ breast cancer progression and invasion, this manuscript emphasizes on the transcriptional regulatory role of super-enhancers (SE). The study reveals an association between TWEAK/fn14 signaling and metabolic alteration. A major conclusion is that TWEAK/fn14 signaling could activate NAMPT expression via a SE-Promoter crosstalk leading to increased NAD⁺ production. Overall, the study is interesting but needs to be improved.

General comments:

Comment B1

1- A major limitation of this investigation relates to the origin of TWEAK which remains elusive from a biological perspective. Considering the complexity of the tumor microenvironment, and previous implication of TWEAK/fn14 signaling stromal cell differentiation processes (e.g., endothelial cells, fibroblasts), a thorough evaluation of TWEAK/fn14 expression in bulk tumors is necessary. Perhaps both TWEAK and fn14 are expressed from the same cells which would suggest an autocrine mechanism.

Response B1

Thank you for pointing this out. We have addressed this by analysing scRNA-seq data from a primary TNBC tumour and its matched lymph node sample where metastasis was detected (1). To which, we noted that in the primary tumour setting, TWEAK is most highly expressed in endothelial cells. In contrast, TWEAK is most highly expressed in monocytes and macrophages in the metastatic setting (Fig. 1,2 (Extended Data Fig. 1e,f in manuscript)). Although epithelial cells do not express much TWEAK, they display the highest expression levels of *Fn14* in both primary and metastatic tumour microenvironments. This analysis suggests that endothelial cells, monocytes and macrophages from the tumour microenvironment are the main sources of TWEAK in their respective settings. Moreover, TWEAK/Fn14 activation in cancer cells is likely to occur through a paracrine mechanism in both primary and metastatic settings.

Comment B2

2- Similarly to TNBCs, Fn14 is also highly expressed in Her2 breast cancer subtypes (Fig.1a). Therefore, it would make sense to evaluate whether the identified mechanisms remain relevant for Her2+ breast cancer. Additionally, how to uncouple the TWEAK/fn14-mediated epigenetic changes from genetic alterations remains a concern.

Response B2

Thank you for the suggestion. To address this, we have examined the Fn14 protein expression levels, TWEAK/Fn14-mediated intracellular signalling pathways, transcriptomic and epigenetic changes in HER2 breast cancer cell lines. Despite the high *Fn14* mRNA expression levels observed in HER2 patients (Figure 3 (Fig. 1a in manuscript)), the protein expression levels of Fn14 in HER2 cell lines were low compared to TNBC cell lines (Fig. 10 (Fig. 1c in manuscript)). This suggests that HER2 breast cancers may have a subtype-specific post-transcriptional regulation of Fn14 which needs to be further evaluated in an independent study.

Further investigation into the TWEAK/Fn14-activated signalling pathways revealed the induction of non-canonical NF- κ B pathway in HER2 breast cancer cell lines upon TWEAK stimulation, whereas the canonical NF- κ B, ERK and JNK pathways remain unchanged (Fig. 11 (Extended Data Fig. 4f in manuscript)). This indicates that TWEAK/Fn14-mediated JNK pathway activation is TNBC specific.

Additional analyses of the TWEAK/Fn14-mediated transcriptomic and epigenetic changes in HER2 breast cancer cell lines compared to TNBC breast cancer cell lines indicate that the identified mechanisms observed in TNBC are absent in HER2 cell lines and therefore subtype-specific (Fig. 12-14 (Fig. 2a, Fig. 3a,d in manuscript, respectively)).

To uncouple the TWEAK/Fn14-mediated epigenetic changes from genetic alterations, we performed a TWEAK washout experiment where the TNBC cell lines, MDA-MB-231 and Hs578T, were treated with TWEAK for 48h, and then cultured in TWEAK-free DMEM media to washout the effects of TWEAK. Here, we demonstrate that TWEAK washout reverses non-canonical NF- κ B and JNK pathway activation (Fig. 15a (Supplementary Fig. 3a in manuscript)). In addition, we observed the reversal of TWEAK/Fn14-driven enhancer changes in TNBC cells (Fig. 15b (Supplementary Fig. 3b in manuscript)). Hence, our data indicates that the TWEAK/Fn14 mediated epigenetic changes detected in TNBC cell lines are not driven by genetic alterations but rather via the intracellular signalling mechanisms triggered during TWEAK/Fn14 activation.

Figure 10. Fn14 protein expression analysed by western blotting in TNBC, HER2 and ER-positive breast cancer cell lines.

Figure 11. NF- κ B and MAPK pathway signalling regulators analysed by western blotting in MDA-MB-231, SKBR3 and MDA-MB-453 cells, treated with and without TWEAK. MDA-MB-231 is added as a reference for comparison.

Figure 12. Heatmap depicting the average expression of TWEAK/Fn14 differentially regulated TNBC genes in MDA-MB-231, Hs578T, SKBR3, MDA-MB-453, MCF7 and CAMA1 cell lines, treated with and without TWEAK. Data shown represent n=3 biological replicates.

Figure 13. ATAC-seq signals of MDA-MB-231, Hs578T, SKBR3 and MCF7 cells, treated with and without TWEAK, as well as the merged signals of Fn14 high TNBC patient tumours and their matched normal samples (2086, 8370, 2963, 6122, 8850 and 5929) at differentially regulated chromatin accessible sites in TNBC cell lines following TWEAK treatment.

Figure 14. H3K27ac ChIP-seq signals of MDA-MB-231, Hs578T, SKBR3, MDA-MB-453, MCF7 and CAMA1 cells, treated with and without TWEAK, as well as the merged signals of Fn14 high TNBC patient tumours and their matched normal samples at differentially regulated enhancer sites in TNBC cell lines.

Figure 15. Wash off ameliorates TWEAK-driven non-canonical NF- κ B pathway, JNK pathway and enhancer activation in TNBC. (a) NF- κ B and JNK pathway signalling regulators analysed by western blotting in untreated, TWEAK-treated and TWEAK-treated with 96h wash-off MDA-MB-231 and Hs578T cells. (b) H3K27ac ChIP-seq signals of untreated, TWEAK-treated and TWEAK-treated with 96h wash-off MDA-MB-231 and Hs578T cells at TWEAK/Fn14-driven differential TNBC enhancer sites.

Specific comments:

Major comments:

Comment B3

1- Fig.2d suggests that similar pathways could be activated in both TNBC and luminal BC. However, the TWEAK/fn14 signaling associates with poor survival outcomes in TNBC only. Therefore, one would wonder if such differences were driven by TNBC-specific signaling pathways as indicated in Fig.2d (e.g., NAD Biosynthesis, MYC, methionine, JNK, cJun). Interestingly, these results also indicate a TWEAK-mediated pathways enrichment in Luminal models. How do you reconcile these cell-specific pathways to tumor progression and overall survival outcomes in breast cancer patients?

Response B3

Thank you for the insightful comment. We have addressed this by extracting the genes associated with the subtype-specific enriched pathways and performed a survival analysis of these genes in the respective subtypes. Here, we found that TNBC and HER2-specific enriched pathways activated by TWEAK/Fn14 signalling are significantly correlated with poorer survival in basal and HER2 patients respectively. These TNBC and HER2-specific pathways have also been reported to drive oncogenic processes including EMT and drug resistance in their respective breast cancer subtypes (9-12). In contrast, Luminal type-specific enriched pathways activated by TWEAK/Fn14 signalling are associated with tumour suppressive functions and improved survival in ER-positive patients (Fig. 16 (Extended Data Fig. 4e in manuscript)) (13-15). These results are consistent with the survival analysis performed based on Fn14 levels (Fig. 17 (Supplementary Fig.1 in manuscript)). Hence, our data indicate that these subtype-specific pathways contribute to the tumour progression and overall survival outcomes of patients with the respective breast cancer subtype.

Figure 16. Kaplan Meier plot depicting the relapse-free survival of Basal, HER2, Luminal A and Luminal B breast cancer patients based on the average expression of genes associated with TWEAK-activated subtype-specific enriched pathways.

Figure 17. Kaplan Meier plot depicting the relapse-free survival of *Fn14* high and low Basal, HER2, Luminal A and Luminal B breast cancer patients.

Comment B4

2- To assess TWEAK/*fn14* signaling, most epigenetic experiments have been based on TWEAK exposure. As *fn14* knockout models have been established and characterized (Sup.Fig1), including more genetic approaches to strengthen the epigenetic and transcriptional role of *fn14* will significantly

strengthen the results. For instance, assessing epigenetic changes using fn14 KO cells +/- TWEAK would be indicated.

Response B4

Thank you for the suggestion. To further highlight the genetic role of Fn14, we have performed H3K27ac ChIP-seq using untreated and TWEAK-treated MDA-MB-231 and Hs578T control and Fn14 KO cells. Here, we observed the lack of TWEAK/Fn14-driven epigenetic changes in Fn14 KO cells following TWEAK exposure relative to control cells (Fig. 18a (Extended Data Fig. 2e in manuscript)). In addition, we found that Fn14 KO abolished TWEAK-mediated non-canonical NF- κ B and JNK activation, in MDA-MB-231 and Hs578T cells (Fig. 18b (Extended Data Fig. 2a in manuscript)), which have been reported to be involved in numerous transcriptional processes (16, 17). This data re-affirms our earlier findings that the TNBC TWEAK/Fn14-driven epigenetic changes are mediated via TWEAK binding to the Fn14 receptor.

Figure 18. (a) H3K27ac ChIP-seq signals of control and Fn14 KO MDA-MB-231 and Hs578T cells, treated with and without TWEAK at differentially regulated enhancer sites in TNBC cell lines. (b) Fn14, NF- κ B and MAPK pathway signalling regulators analysed by western blotting in control and Fn14 KO MDA-MB-231 and Hs578T cells, treated with and without TWEAK.

Comment B5

3- The authors claim that NAMPT is specifically induced in TNBC but not in Luminal models (Fig6 a,b, and e). For such claims to be true based on western blot experiments, a similar amount of protein must be loaded for each sample. It also remains possible that the lack of expression in luminal cells is related to genomic differences (e.g., mutations). This needs further evaluation. Contrasting with the TNBC models, CAMA1 shows a decreased level of NAMPT following TWEAK treatment. How do the authors interpret this result?

Response B5

To validate our claim, we have performed western blot densitometric analyses. Through this, we verified that TWEAK stimulation induces NAMPT expression specifically in TNBC but not in HER2 and ER-positive cell lines (Fig. 19 a-c (Supplementary Fig. 7a, b, c in manuscript)).

In the TNBC models, we observe high NAMPT expression which has been reported to be associated with the absence of ER (18). The NAMPT levels in TNBC are further upregulated by TWEAK stimulation and this correlates with the presence of the *NAMPT* SE and its upregulation following TWEAK stimulation. Given the absence of the *NAMPT* SE and low NAMPT expression in other subtypes, it suggests that the *NAMPT* SE may be responsible for regulating NAMPT expression (Fig. 20 (Fig. 6c in manuscript)).

It is interesting that TWEAK stimulation downregulates NAMPT expression in the CAMA1 cell line (Fig. 21 (Fig. 6a in manuscript)). Closer inspection of the enhancer signatures of CAMA1 at the *NAMPT* locus shows a decrease in *NAMPT* enhancer #1 (Fig. 20 (Fig. 6c in manuscript)) which could result in the downregulation of NAMPT expression. However to confirm this, further investigation in a separate study is required.

Figure 19. Relative NAMPT expression to GAPDH, (a) normalised to each cell line's untreated sample in Figure 6a, (b) normalised to each tumour sample's matched normal tissue in Figure 6b and (c) normalised to the control untreated in Figure 6d.

Figure 20. *NAMPT* locus. Top: H3K27ac HiChIP heatmap of TWEAK-treated MDA-MB-231 cells. Middle: H3K27ac ChIP-seq of untreated and TWEAK-treated MDA-MB-231, Hs578T, SKBR3, MDA-MB-453, MCF7 and CAMA1 cells. Bottom: chromatin interactions in untreated and TWEAK-treated MDA-MB-231 cells, and merged H3K27ac ChIP-seq of Fn14 high TNBC tumours and their matched normal samples (2086, 8370, 2963, 6122, 8850 and 5929).

Figure 21. Western blot analysis of NAMPT expression in untreated and TWEAK-treated TNBC, HER2 and ER-positive breast cancer cell lines

Comment B6

4- The ChIP-PCR results suggest that MAFG mediates TWEAK/fn14 signaling at NAMPT enhancer #3. However, it remains unclear why only enhancer #3 is impacting NAMPT transcription and no other enhancers. It seems adequate to also evaluate the impact of MAFG on Enhancers #1 and #2. Moreover, assuming that NAMPT upregulation is specific to TNBC, the expression and binding of MAFG should be evaluated in other breast cancer subtypes. This experiment is determinant considering that TWEAK treatment appears to increase fn14 level in luminal breast cancer as well (see Fig.2e).

Response B6

Thank you for the interesting comment. To evaluate the impact of MAFG on the enhancers within the *NAMPT* locus, we further performed H3K27ac ChIP-qPCR at *NAMPT* enhancer #1 and #2 in control and shMAFG MDA-MB-231 and Hs578T cells. Here, we demonstrate that with TWEAK stimulation, *NAMPT* enhancers #1 and #2 are still significantly activated in control and shMAFG cells (Fig. 9b,c (Extended Data Fig. 8b, c in manuscript)). In contrast, *NAMPT* enhancer #3 activation was abrogated in shMAFG cells, indicating that MAFG specifically impacts *NAMPT* enhancer #3 and mediates *NAMPT* expression via *NAMPT* enhancer #3.

We further evaluated the expression of MAFG in our breast cancer cell lines and observed higher MAFG levels in the TNBC cell lines (Fig. 22 (Extended Data Fig. 7c in manuscript)). We also performed MAFG ChIP-seq in the breast cancer cell lines, which further confirmed the binding of MAFG to *NAMPT* enhancer #3 selectively in TNBC cells (Fig. 8 (Extended Data Fig.7f in manuscript)). Altogether, our data suggests that MAFG is more highly expressed in TNBC cell lines and provides evidence for the specific binding of MAFG to the *NAMPT* enhancer #3 only in TNBCs to drive TWEAK/Fn14-mediated *NAMPT* expression.

Figure 22. MAFG protein expression analysed by western blotting in MDA-MB-231, Hs578T, SKBR3, MDA-MB-453, MCF7, T47D and CAMA1.

Comment B7

5- The overexpression of TWEAK in MCF7 cells shows a decreased trend in primary tumor proliferation which may indicate an opposite function of TWEAK/fn14 signaling in luminal breast cancer models when compared to TNBC. Unfortunately, only 3 mice were included in each group (Sup. Fig.2b-c). Including more mice would help strengthen the conclusions.

Response B7

Thank you for the suggestion. We have included two additional mice per group in the revised manuscript, further confirming our earlier observations that TWEAK overexpression does not significantly affect primary tumour growth (Fig. 23 (Supplementary Fig. 2b,c in manuscript)).

Figure 23. Persistent Fn14 activation promotes tumour growth in TNBC but not ER-positive breast cancer cells. (a) Images of tumours extracted from mice 8 weeks after being injected with MCF7 cells (n=5 biological replicates) harbouring the overexpression control or TWEAK overexpression. (b) Tumour growth plot depicts the weekly average tumour volume (mean \pm s.d) from mice injected with MCF7 cells overexpressing luciferase and overexpression control or TWEAK. T-test was used for statistical analysis. There was no tumour growth in the 5th MCF7 TWEAK Overexpression mouse. T-test was used for statistical analysis.

Comment B8

6- In Fig.6j, it is unclear how the metastatic sites were defined and quantified. Is this based on ex vivo imaging? Additionally, including representative metastatic lesions is required to ascertain the origin of bioluminescence signals. Assessing differences at the primary site may also be relevant.

Response B8

The metastatic sites were quantified by using the ROI tool in the Living Imaging Software (Perkin Elmer) to detect the bioluminescent signal across the entire mouse and we subtracted the bioluminescent signal from the primary tumour. This information has been added to the materials and methods section for clarity.

The images were acquired based on live animal IVIS imaging and therefore no representative metastatic lesions could be provided.

Comment B9

7- Fig.6d and Fig.6e: A more quantitative approach with statistics would be beneficial.

Response B9

Thank you for the comment. To address this, we have performed additional western blot densitometric analyses (Fig. 24) Supplementary Fig. 7d-g in manuscript)).

Figure 24. Western blot densitometries. Relative (a) MAFG and (b) NAMPT expression to GAPDH, normalised to the control untreated MDA-MB-231 sample in Extended Data Figure 8. Relative (c) MAFG and (d) NAMPT expression to GAPDH, normalised to the control untreated Hs578T sample in Extended Data Figure 8. T-test was used for in Tumour vs Normal statistical analysis while Two-way ANOVA was used for statistical analysis in all other plots, where *P < 0.05; **P < 0.01; ***P < 0.001.

Comment B10

8- The enhanced AP-1 binding activity in TNBC following TWEAK exposure suggests that AP-1 mediates TWEAK/fn14 signaling in these models. These results require further functional validation to determine whether the impact of TWEAK/tnf14 on cell proliferation, migration, invasion, and overall metastasis *in vivo* is truly mediated by AP1 signaling. Additionally, it is unclear how MAFG is recruited at enhancer sites following TWEAK/fn14 activation. Is it known whether NFKB could mediate this process? Also, assessing the association (correlation) between MAFG and NAMPT from patient datasets would strengthen the clinical relevance of the mechanism.

Response B10

Thank you for the suggestion. To address this comment, we have performed additional *in vitro* and *in vivo* experiments using the AP-1 inhibitor T5224. Here, we demonstrate that AP-1 inhibition abolished the proliferation and invasion of TNBC cells *in vitro*. Consistent with the *in vitro* data, T5224 supplementation reduced the metastasis of TNBC cells *in vivo*, without affecting tumour growth (Fig. 25 (Extended Data Fig. 6 in manuscript)). These results were also observed in head and neck squamous carcinoma where T5224 inhibited metastasis *in vitro* and *in vivo* (19). It is possible that the lack of tumour growth inhibition *in vivo* compared to the anti-proliferative effects observed *in vitro* may

be attributed to tumour microenvironment factors, such as crosstalk with adipocytes and the adipose stroma, reported previously (20-23). Altogether, this suggests that AP-1 factors are critical regulators of TWEAK/Fn14-induced oncogenic SEs and function.

To elucidate how MAFG is recruited at enhancer sites and whether NF κ B has a role in this, we have performed MAFG and NF κ B2/p52 ChIP-seq in untreated and TWEAK-treated MDA-MB231 cells. In the revised manuscript, we compared the MAFG, NF κ B2/p52, H3K27ac ChIP-seq peaks and ATAC-seq signals of MDA-MB-231 cells at TWEAK/Fn14-driven differential TNBC H3K27ac regions. Here, we show that upon TWEAK treatment, there is increased chromatin accessibility at TWEAK/Fn14-driven TNBC upregulated H3K27ac sites in TWEAK-treated versus untreated TNBC cells and this is associated with increased MAFG binding. In contrast, there was little change in MAFG binding at downregulated enhancer sites. Additionally, we noted the weak NF κ B2/p52 ChIP-seq signals at all MAFG bound enhancer sites, suggesting that NF κ B2/p52 is not crucial for TWEAK/Fn14-driven MAFG recruitment to enhancers (Fig. 26 (Extended Data Fig. 7d in manuscript)). Taken together, this suggests that MAFG recruitment at enhancer sites following TWEAK/Fn14 activation is mediated by increased chromatin accessibility.

We further examined the association between *MAFG* and *NAMPT* expression utilising the TCGA clinical datasets. Here we found that in both the TCGA BRCA and TCGA PanCancer datasets, there is a positive correlation between *MAFG* and *NAMPT* expression (Fig. 27 (Extended Data Fig. 7e in manuscript)), further highlighting the clinical relevance of the described mechanism.

Figure 25. (a) Proliferation assay plot depicts the average number of cells counted relative to 0h (mean \pm s.d) in untreated, TWEAK-treated and TWEAK + 10 μ M T5224 treated MDA-MB-231, Hs578T, n=3 biological replicates. (b) Transwell invasion assay was performed in untreated, TWEAK-treated and TWEAK + 10 μ M T5224 treated MDA-MB-231, Hs578T. Representative images from n=3 biological replicates are shown. Scale bars: 400 μ m. (c) Plot depicts the average number of invaded cells/frame (mean \pm s.d) from n=3 biological replicates, across four fields per replicate. (d) Images of tumours extracted from mice 8 weeks after being injected with luciferase and TWEAK overexpressing MDA-MB-231 cells and treated with vehicle or 10 μ M T5224 (n=5 biological replicates). (e) Tumour growth plot depicts the weekly average tumour volume (mean \pm s.d) from mice injected with MDA-MB-231 cells overexpressing luciferase and TWEAK, treated with vehicle or 150mg/kg T5224. (f) IVIS tracking of mice injected with MDA-MB-231 cells overexpressing luciferase and TWEAK, treated with vehicle or 150mg/kg T5224. Representative bioluminescent images of the animals were taken at 1 and 8 weeks after orthotopic xenograft. (g) Plot depicts total flux at the metastatic sites of each animal

after 8 weeks in n=5 biological replicates. Two-way ANOVA was used for statistical analysis in *in vitro* proliferation and invasion assay. T-test was used for statistical analysis in *in vivo* assays. *P < 0.05; **P < 0.01; ***P < 0.001.

Figure 26. MAFG, p52, and H3K27ac ChIP-seq, and ATAC-seq signals of MDA-MB-231 cells treated with and without TWEAK at TWEAK/Fn14-driven differential TNBC H3K27ac peaks.

Figure 27. Correlation between *MAFG* and *NAMPT* expression in the TCGA BRCA and TCGA PanCancer RNA-seq dataset.

Minor Comments:

Comment B11

9- In Sup. Fig.3, the survival follow-up stops at month 60 while other breast cancer subtypes reach month 120. Is there any particular reason for such variation?

Response B11

Thank you for pointing this out. It was an error which we have rectified in the revised manuscript. The survival follow-up period for basal patients has been changed to 120 months as well (Fig. 28 (Supplementary Fig. 1 in manuscript)).

Figure 28. High Fn14 expression confers worse survival in Basal-like and HER2 patients. Kaplan Meier plot depicting the relapse-free survival of *Fn14* high and low Basal, HER2, Luminal A and Luminal B breast cancer patients.

Reviewer C Comments

REVIEWER#3

Comment C1

In the manuscript "TWEAK/Fn14 signalling driven super-enhancer reprogramming promotes pro-metastatic metabolic rewiring in Triple-Negative Breast Cancer" the authors propose that expression of Fn14 is a potential driver of epigenetic and metabolic reprogramming in Triple Negative Breast Cancer (TNBC). This is an interesting manuscript and may have potential clinical significance. However, there are several significant pitfalls that need to be resolved before the conclusions of the manuscript can be supported by the data.

Major: The main issue with the manuscript is the paucity of clinical data sets. The manuscript is heavily dependent on the use of breast cancer cell lines in vitro. The use of avatars and also of immune competent mouse models is imperative to confirm the conclusions.

Cell lines may not reflect the clinical cancer and its microenvironment. Therefore, at this point the conclusions are based on cultured cells.

Response C1

Thank you for the comments. To further highlight the clinical relevance of our findings, we have included scRNA-seq data from a primary TNBC tumour and matched lymph node metastases to elucidate the source of TWEAK in the primary tumour and metastatic setting of TNBC patients. Our analysis revealed that in both the primary and metastatic setting, *Fn14* was most highly expressed in epithelial cells. Whereas *TWEAK* was most highly expressed in endothelial cells in the primary setting

and in monocytes and macrophages in the metastatic setting. This suggests that TWEAK/Fn14 activation in TNBC cells occurs in a paracrine fashion (Fig. 29 (Extended Data Fig. 1 in manuscript)). We have also incorporated the TCGA BRCA and TCGA PanCancer RNA-seq datasets to illustrate the positive correlation between *MAFG* and *NAMPT* expression levels in different cancer patients (Fig. 27 (Extended Data Fig.7e in manuscript)), further highlighting the clinical relevance of our findings.

To assess the role of the tumour microenvironment during TWEAK/Fn14 signalling, we have performed orthotopic tumour xenografts using wild-type, immune competent BALB/cN mice. The mice were engrafted with luciferase overexpressing 4T1 cells (mouse TNBC cell line) harbouring either the CRISPR control or mouse *Fn14* receptor CRISPR knockdown (KD). Here, we demonstrate that in the immune competent mice, KD of *Fn14* expression downregulates tumour growth and metastasis (Fig. 30 (Extended Data Fig. 3 in manuscript)). These results are consistent with our observations using TNBC cell lines, immune-deficient mice and scRNA-seq data, validating the significant role of TWEAK/Fn14 signalling in TNBC progression and metastasis.

The main focus of this study is to elucidate the mechanistic role of the TWEAK/Fn14 signalling pathway in TNBC progression via super-enhancer reprogramming and not identifying new drug treatments for TNBC. We agree that the use of avatar mice in future studies will be beneficial to test the efficacy of existing or novel therapeutic drugs targeting Fn14 or its downstream targets. However, this is a separate study that is beyond the scope of our current work.

Figure 29. scRNA-seq analyses of primary TNBC tumour and its lymph node metastases reveal that *Fn14* is mainly expressed in breast TNBC epithelial cells while *TWEAK* is mainly expressed in endothelial cells and macrophages. UMAP overview of cells from primary TNBC tumour and its matched lymph nodes with metastasis detected, analysed by scRNA-seq, acquired from (1) depicting (a) the origin of each cell, (b) the cell type, and the expression of (c) *Fn14* and (d) *TWEAK* of each cell. Plot depicting the average (e) *Fn14* and (f) *TWEAK* expression of single cells in the primary TNBC tumour and metastasis positive lymph node samples. T-test was used for statistical analysis comparing epithelial cells or endothelial cells versus all other cell types, monocytes versus all other cell types except macrophages and macrophages versus all other cell types except monocytes. *P < 0.05; **P < 0.01; ***P < 0.001.

Figure 30. (a) Images of tumours extracted from mice 42 days after being injected with luciferase overexpressing 4T1 cells harbouring either CRISPR control or *Fn14* KD (n=5 biological replicates). (b) Tumour growth plot depicts the weekly average tumour volume (mean \pm s.d) from mice injected with 4T1 cells overexpressing luciferase and harbouring either CRISPR control or *Fn14* KD. (c) IVIS tracking of mice injected with 4T1 cells overexpressing luciferase and harbouring either CRISPR control or *Fn14* KD. Representative bioluminescent images of the animals were taken at days 3 and 42 after orthotopic xenograft. (d) Plot depicts total flux at the metastatic sites of each animal after 42 days in n=5 biological replicates. T-test was used for statistical analysis in *in vivo* assays. *P < 0.05; **P < 0.01; ***P < 0.001.

Comment C2

It is also important to highlight that culture conditions of these cells in often very artificial and may significantly change the outcome of metabolism and its epigenetic conditions. Thus, the validation of the data is clinically relevant data sets is imperative.

Response C2

We agree that the culture conditions of cells are artificial and may not accurately reflect the metabolic processes and epigenetic conditions in patients. As such, we have attempted to illustrate the clinical epigenetic conditions through the use of 6 *Fn14* high TNBC tumour samples and their matched normal samples (Fig. 13, 14 (Fig. 3 a,d in manuscript), Fig. 31 (Extended Data Fig. 5 in manuscript)). To evaluate the metabolic processes that occur in patients, we have utilised 3 *Fn14* high TNBC tumours and their matched normal samples (samples 2086, 6122 and 8370) to perform ATP and NAD/NADH assays. Here, we find that in all tumour samples, the levels of ATP and NAD/NADH were elevated compared to their matched normal samples (Fig. 32 (Extended Data Fig. 9 c,d in manuscript)). These findings are consistent with our results from cell lines and indicates that similar metabolic mechanisms also occur in TNBC patients, highlighting its clinical relevance.

Figure 31. ATAC-seq signals of MDA-MB-231, Hs578T, SKBR3 and MCF7 with and without TWEAK, as well as the merged signal of Fn14 high TNBC patient tumours and their matched normal samples (2086, 8370, 2963, 6122, 8850 and 5929) at differentially regulated chromatin accessible sites in (a) MCF7 and (b) SKBR3 cells following TWEAK treatment. (b) H3K27ac ChIP-seq signals of MDA-MB-231, Hs578T, SKBR3, MDA-MB-453, MCF7 and CAMA1 with and without TWEAK, as well as the merged signal of Fn14 high TNBC patient tumours and their matched normal samples at differentially regulated enhancer sites in the (c) ER-positive and (d) HER2 cell lines.

Figure 32. (a) NAD⁺/NADH and (b) intracellular ATP levels in Fn14 and NAMPT high TNBC tumours and their matched normal samples (2086, 6122 and 8370). T-test was used for statistical analysis in all assays. *P < 0.05; **P < 0.01; ***P < 0.001.

References

- Xu K, Wang R, Xie H, Hu L, Wang C, Xu J, et al. Single-cell RNA sequencing reveals cell heterogeneity and transcriptome profile of breast cancer lymph node metastasis. *Oncogenesis*. 2021;10(10):66.
- Saitoh T, Nakayama M, Nakano H, Yagita H, Yamamoto N, Yamaoka S. TWEAK induces NF-kappaB2 p100 processing and long lasting NF-kappaB activation. *J Biol Chem*. 2003;278(38):36005-12.
- Varfolomeev E, Goncharov T, Maecker H, Zobel K, Komuves LG, Deshayes K, et al. Cellular inhibitors of apoptosis are global regulators of NF-kappaB and MAPK activation by members of the TNF family of receptors. *Sci Signal*. 2012;5(216):ra22.
- Cherry EM, Lee DW, Jung JU, Sitcheran R. Tumor necrosis factor-like weak inducer of apoptosis (TWEAK) promotes glioma cell invasion through induction of NF-kappaB-inducing kinase (NIK) and noncanonical NF-kappaB signaling. *Mol Cancer*. 2015;14:9.
- Willis AL, Tran NL, Chatigny JM, Charlton N, Vu H, Brown SA, et al. The fibroblast growth factor-inducible 14 receptor is highly expressed in HER2-positive breast tumors and regulates breast cancer cell invasive capacity. *Mol Cancer Res*. 2008;6(5):725-34.
- Tran NL, McDonough WS, Donohue PJ, Winkles JA, Berens TJ, Ross KR, et al. The human Fn14 receptor gene is up-regulated in migrating glioma cells in vitro and overexpressed in advanced glial tumors. *Am J Pathol*. 2003;162(4):1313-21.
- Yin J, Liu YN, Tillman H, Barrett B, Hewitt S, Ylaya K, et al. AR-regulated TWEAK-FN14 pathway promotes prostate cancer bone metastasis. *Cancer Res*. 2014;74(16):4306-17.
- Chao DT, Su M, Tanlimco S, Sho M, Choi D, Fox M, et al. Expression of TweakR in breast cancer and preclinical activity of enavatuzumab, a humanized anti-TweakR mAb. *J Cancer Res Clin Oncol*. 2013;139(2):315-25.
- Harper KL, Sosa MS, Entenberg D, Hosseini H, Cheung JF, Nobre R, et al. Mechanism of early dissemination and metastasis in Her2(+) mammary cancer. *Nature*. 2016;540(7634):588-92.
- Kang HJ, Yi YW, Hong YB, Kim HJ, Jang YJ, Seong YS, et al. HER2 confers drug resistance of human breast cancer cells through activation of NRF2 by direct interaction. *Sci Rep*. 2014;4:7201.
- Butler M, van der Meer LT, van Leeuwen FN. Amino Acid Depletion Therapies: Starving Cancer Cells to Death. *Trends Endocrinol Metab*. 2021;32(6):367-81.
- Zimmerli D, Brambillasca CS, Talens F, Bhin J, Linstra R, Romanens L, et al. MYC promotes immune-suppression in triple-negative breast cancer via inhibition of interferon signaling. *Nat Commun*. 2022;13(1):6579.
- Jin L, Zhang J, Fu HQ, Zhang X, Pan YL. FOXO3a inhibits the EMT and metastasis of breast cancer by regulating TWIST-1 mediated miR-10b/CADM2 axis. *Transl Oncol*. 2021;14(7):101096.

14. Dansen TB, Burgering BM. Unravelling the tumor-suppressive functions of FOXO proteins. *Trends Cell Biol.* 2008;18(9):421-9.
15. Kim GC, Lee CG, Verma R, Rudra D, Kim T, Kang K, et al. ETS1 Suppresses Tumorigenesis of Human Breast Cancer via Trans-Activation of Canonical Tumor Suppressor Genes. *Front Oncol.* 2020;10:642.
16. Ghosh G, Wang VY-FJFC. Origin of the functional distinctiveness of NF- κ B/p52. *Frontiers in Cell and Developmental Biology.* 2021:3338.
17. Sehgal V, Ram PT. Network Motifs in JNK Signaling. *Genes Cancer.* 2013;4(9-10):409-13.
18. Zhou SJ, Bi TQ, Qin CX, Yang XQ, Pang K. Expression of NAMPT is associated with breast invasive ductal carcinoma development and prognosis. *Oncol Lett.* 2018;15(5):6648-54.
19. Kamide D, Yamashita T, Araki K, Tomifuji M, Tanaka Y, Tanaka S, et al. Selective activator protein-1 inhibitor T-5224 prevents lymph node metastasis in an oral cancer model. *Cancer Sci.* 2016;107(5):666-73.
20. Park J, Euhus DM, Scherer PE. Paracrine and endocrine effects of adipose tissue on cancer development and progression. *Endocr Rev.* 2011;32(4):550-70.
21. Romero IL, Mukherjee A, Kenny HA, Litchfield LM, Lengyel E. Molecular pathways: trafficking of metabolic resources in the tumor microenvironment. *Clin Cancer Res.* 2015;21(4):680-6.
22. Nieman KM, Romero IL, Van Houten B, Lengyel E. Adipose tissue and adipocytes support tumorigenesis and metastasis. *Biochim Biophys Acta.* 2013;1831(10):1533-41.
23. Ehemann C, Henley SJ, Ballard-Barbash R, Jacobs EJ, Schymura MJ, Noone AM, et al. Annual Report to the Nation on the status of cancer, 1975-2008, featuring cancers associated with excess weight and lack of sufficient physical activity. *Cancer.* 2012;118(9):2338-66.

REVIEWER COMMENTS

Reviewer #1 (Remarks to the Author):

The authors have provided appropriate responses to my initial of review and significantly improved the manuscript. I don't have significant additional concerns.

Reviewer #2 (Remarks to the Author):

The revised manuscript has significantly improved in terms of quality and rigor. The additional ATAC-seq and ChIP experiments have certainly strengthened the proposed mechanistic of action of TWEAK/Fn14 in TNBC. While a few major concerns were the (i) restriction of the TWEAK/fn14 signaling to TNBC and (ii) the source of TWEAK, (iii) and its clinical relevance, the provided responses seem appropriate. This may grant publication.

Reviewer #3 (Remarks to the Author):

In the revised manuscript the authors show new data on single cell RNAseq that indicates that Fn14 is mostly expressed in endothelial cells and that TWEAK is mostly expressed in either endothelial cells or immune cells. It is not clear how that correlates with the tumor data. Sre the tumor cells or normal epithelial cells expressing Fn14. These could very significantly change the interpretation of the study. More importantly since protein level is what really matters, I would strongly recommend that the authors include protein analysis of the microenvironment including the tumor cells (in native tumors not cell line) for these proteins.

Response to Reviewer #3

Comment C1:

In the revised manuscript the authors show new data on single cell RNAseq that indicates that Fn14 is mostly expressed in endothelial cells and that TWEAK is mostly expressed in either endothelial cells or immune cells. It is not clear how that correlates with the tumor data.

Response C1:

In our revised manuscript, we have utilised single cell RNA-seq data from TNBC tumour and matched metastasis-positive lymph node samples for the analysis of Fn14 and TWEAK expression. Our analysis showed that Fn14 is most highly expressed in tumour epithelial cells. In contrast, TWEAK is mainly expressed in endothelial cells in primary tumours, and monocytes and macrophages in metastatic samples. This analysis was performed to dissect our bulk tumour data, demonstrating that cancer cells mainly express Fn14 and are the main contributors of Fn14 RNA and protein expression observed in the TCGA BRCA RNAseq dataset (Fig. 1 (Fig. 1a in manuscript)) and matched TNBC tumour western blot, respectively (Fig. 2 (Fig. 1b in manuscript)). In addition, this data suggests that TWEAK/Fn14 signalling in TNBC cells occurs through a paracrine mechanism.

These results were further supported by the orthotopic engraftment of 4T1 (mouse TNBC cell line of epithelial origin) cells in immune competent mice whereby knockdown of *Fn14* significantly reduced tumour growth and metastasis (Fig. 3 (Extended Data Fig. 3 in manuscript)).

Figure 1. Relative *Fn14* gene expression levels in Basal-like, HER2, Luminal A, Luminal B, Normal-like and non-cancer patient samples from the TCGA BRCA RNA-seq dataset.

Figure 2. Fn14 protein expression analysed by western blotting in (b) TNBC patient tumours (T) and their matched normal samples (N).

Figure 3. (a) Images of tumours extracted from mice 42 days after being injected with luciferase overexpressing 4T1 cells harbouring either CRISPR control or *Fn14* KD (n=5 biological replicates). (b) Tumour growth plot depicts the weekly average tumour volume (mean \pm s.d) from mice injected with 4T1 cells overexpressing luciferase and harbouring either CRISPR control or *Fn14* KD. (c) IVIS tracking of mice injected with 4T1 cells overexpressing luciferase and harbouring either CRISPR control or *Fn14* KD. Representative bioluminescent images of the animals were taken at days 3 and 42 after orthotopic xenograft. (d) Plot depicts total flux at the metastatic sites of each animal after 42 days in n=5 biological replicates. T-test was used for statistical analysis in *in vivo* assays. *P < 0.05; **P < 0.01; ***P < 0.001.

Comment C2:

Are the tumor cells or normal epithelial cells expressing Fn14. These could very significantly change the interpretation of the study. More importantly since protein level is what really matters, I would strongly recommend that the authors include protein analysis of the microenvironment including the tumor cells (in native tumors not cell line) for these proteins.

Response C2:

Both tumour cells and normal epithelial cells express Fn14. However, tumour cells express higher levels of Fn14 as demonstrated through analysis of the TCGA BRCA RNAseq dataset (Fig. 1 (Fig. 1a in manuscript)) and western blot analysis of matched TNBC tumour sets (Fig. 2 (Fig. 1b in manuscript)). It should be noted that these experiments utilised tumour samples with their microenvironments.

Through scRNA-seq analysis of matched TNBC tumour and metastasis positive lymph node samples, we find that Fn14 is most highly expressed in cancer epithelial cells whereas TWEAK is mainly expressed in endothelial cells in primary tumours, and monocytes and macrophages in metastatic samples. To correlate this with protein expression, we analysed bulk breast cancer tumour datasets (tumours also containing cells from the microenvironment) where both RNA-seq and proteomic analyses were performed on the same tumours (1, 2). Here, we observed a significantly positive correlation between the protein and RNA expression of Fn14 and TWEAK (Fig. 4). This supports our data that cells within breast cancer tumours with high *Fn14* RNA expression, display high Fn14 protein levels. Similarly, cells that exhibit high *TWEAK* RNA expression, typically express high levels of TWEAK protein. Hence, taken together with the scRNA-seq results, our data indicates that Fn14 is mostly expressed in cancer epithelial cells within TNBC tumours whilst TWEAK is mostly expressed in endothelial cells, monocytes and macrophages from the tumour microenvironment.

Figure 4. Plots depicting the Spearman correlation of (a, b) Fn14 and (c) TWEAK RNA expression against protein expression from breast cancer patient bulk tumour datasets (1, 2).

References

1. Krug K, Jaehnig EJ, Satpathy S, Blumenberg L, Karpova A, Anurag M, et al. Proteogenomic Landscape of Breast Cancer Tumorigenesis and Targeted Therapy. *Cell*. 2020;183(5):1436-56 e31.
2. Vasaikar SV, Straub P, Wang J, Zhang B. LinkedOmics: analyzing multi-omics data within and across 32 cancer types. *Nucleic Acids Res*. 2018;46(D1):D956-D63.

REVIEWERS' COMMENTS

Reviewer #3 (Remarks to the Author):

The response to my previous comments were not extremely convincing. The n of some experiments continues to be low. For example the metastasis figure is not convincing and appears to be significant despite an enormous variation on the samples with only 2 having a clear increase in metastasis , an outlier analysis would possibly demonstrate this. In general the authors did not approach my questions with data and in a convincing way.

Reviewer #3 (Remarks to the Author):

The response to my previous comments were not extremely convincing. The n of some experiments continues to be low. For example the metastasis figure is not convincing and appears to be significant despite an enormous variation on the samples with only 2 having a clear increase in metastasis, an outlier analysis would possibly demonstrate this. In general the authors did not approach my questions with data and in a convincing way.

Response:

As suggested, we have revised Figure 3D (Figure 1 (Supplementary Figure 5 in manuscript)) to present the data binomially. Here, we observe that *Fn14* KD significantly reduces tumour growth and resulted in a decreasing trend in the incidence of metastasis.

Figure 1. (a) Images of tumours extracted from mice 42 days after being injected with luciferase overexpressing 4T1 cells harbouring either CRISPR control or *Fn14* KD (n=5 biological replicates). (b) Tumour growth plot depicts the weekly average tumour volume (mean \pm s.d) from mice injected with 4T1 cells overexpressing luciferase and harbouring either CRISPR control or *Fn14* KD. (c) IVIS tracking of mice injected with 4T1 cells overexpressing luciferase and harbouring either CRISPR control or *Fn14* KD. Representative bioluminescent images of the animals were taken at days 3 and 42 after orthotopic xenograft. (d) Plot depicts the number of metastatic events observed in animals after 42 days in n=5 biological replicates. Data presented as binomial. Metastasis detected in control mice #2 and #4. Two-sided t-test was used for statistical analysis in tumour volume growth assay. *P < 0.05; **P < 0.01; ***P < 0.001. Source data are provided as a Source Data file.